# LIN-12/Notch signaling instructs postsynaptic muscle arm development by regulating UNC-40/DCC and MADD-2 in *Caenorhabditis elegans*

**Pengpeng Li[1], Kevin M Collins[2], Michael R Koelle[2], Kang Shen[1]\***

[1]Department of Biology, Howard Hughes Medical Institute, Stanford University, Stanford, United States; [2]Department of Molecular Biophysics and Biochemistry, Yale University, New Haven, United States

**Abstract** The diverse cell types and the precise synaptic connectivity between them are the cardinal features of the nervous system. Little is known about how cell fate diversification is linked to synaptic target choices. Here we investigate how presynaptic neurons select one type of muscles, vm2, as a synaptic target and form synapses on its dendritic spine-like muscle arms. We found that the Notch-Delta pathway was required to distinguish target from non-target muscles. APX-1/Delta acts in surrounding cells including the non-target vm1 to activate LIN-12/Notch in the target vm2. LIN-12 functions cell-autonomously to up-regulate the expression of UNC-40/DCC and MADD-2 in vm2, which in turn function together to promote muscle arm formation and guidance. Ectopic expression of UNC-40/DCC in non-target vm1 muscle is sufficient to induce muscle arm extension from these cells. Therefore, the LIN-12/Notch signaling specifies target selection by selectively up-regulating guidance molecules and forming muscle arms in target cells.

**\*For correspondence:** kangshen@stanford.edu

**Competing interests:** The authors declare that no competing interests exist

**Reviewing editor**: Graeme Davis, University of California, San Francisco, United States

## Introduction

Functional neural circuits are generated through coordinated events during the development of the nervous system including cell type specification, neuronal process formation and synaptogenesis. Many studies have demonstrated that stereotyped wiring exists between different cell types. Insights from studies in spinal cord and neocortex development strongly suggest that a combinatorial code of transcription factors mediates cell specification and defines cellular identities among different cells (*Jessell, 2000*; *Shirasaki and Pfaff, 2002*). An emerging literature indicates that precise synaptic connections are specified by diverse molecular mechanisms. Both positive and negative regulators of synapse formation can specify local synaptic connectivity (*Williams et al., 2010*; *Maeder and Shen, 2011*). Homotypic and heterotypic adhesion molecules can determine the synaptic lamina formation and even synaptic partner choice (*Yamagata et al., 2003*; *de Wit et al., 2011*). For example, in *Drosophila*, many transmembrane molecules, including semaphorin, capricious and teneurins, have been suggested to guide synaptic target selection between olfactory receptor neurons (ORN) and projection neurons (PN) (*Komiyama et al., 2007*; *Hong et al., 2009, 2012*). In vertebrate spinal cord, a semaphorin-plexin pathway regulates the connection specificity in the sensory motor circuit (*Pecho-Vrieseling et al., 2009*). In vertebrate retina, retinal ganglion cells (RGCs) form synapses with retinal interneurons in the inner plexiform layer (IPL), generating several synaptic laminae (*Sanes and Yamagata, 2009*). This laminar specificity is directed by both the homophilic interactions of several immunoglobin superfamily (IgSF) proteins, including Sidekicks and Dscam, as well as the inhibitory action of semaphorins (*Yamagata et al., 2002*; *Yamagata and Sanes, 2008*; *Matsuoka et al., 2011*).

**eLife digest** The development of the nervous system involves the formation of complex networks of connections between diverse cell types, such as motor neurons, interneurons and pyramidal cells. However, the mechanisms by which individual cells are programmed to acquire particular identities, and how they are instructed to form connections with other specific cells, remain unclear.

In many species, the Notch signaling pathway has a role in setting up these networks. Notch is a transmembrane protein, which means that it has one component inside the cell and another outside. When a ligand binds to the extracellular part of Notch, this causes the receptor to break in two. The intracellular domain then travels to the nucleus where it can influence gene expression.

The nematode worm (*C. elegans*), which has two Notch receptors, is often used to study the formation of neuronal networks because each worm has only around 300 neurons, and they are connected in roughly the same way in each worm. *C. elegans* relies on two types of cell that are very similar to each other—type-1 and type-2 vulval muscle cells—to lay eggs, and the neurons that trigger egg-laying form synaptic connections on specialized structures called muscle arms. However, these structures are found only in type-2 vulval muscle.

To investigate the mechanisms underlying the formation of the egg-laying circuit, Li et al. screened large numbers of mutant worms to find animals that lacked muscle arms. They identified a number of such mutants, which laid fewer eggs compared to wild-type worms, and found that they all had mutations in genes that encode for proteins or ligands that are involved in the LIN-12/Notch pathway. This pathway mediates cell–cell interactions that help to specify cell fates.

Li et al. showed that type-2 vulval muscle cells develop muscle arms when their neighbors—type-1 vulval muscle cells and vulval epithelial cells—produce enough ligand to activate the LIN-12 Notch receptor on the type-2 vulval muscle cells. They also identified two of the downstream targets of LIN-12, and found that artificially expressing one of these in type-1 vulval muscle cells is sufficient to trigger the formation of muscle arms.

The work of Li et al. provides further evidence that the Notch signalling pathway, which is well known for its role in early development, also acts at later developmental stages to determine cell fate and patterns of connectivity.

Similarly, in *C. elegans*, the heterophilic interaction of two IgSF proteins, SYG-1 and SYG-2, guides the localization of the en passant synapse formation of HSN neurons (*Shen and Bargmann, 2003*; *Shen et al., 2004*). Another IgSF protein UNC-40/DCC, the receptor of UNC-6/Netrin, has been shown to regulate axon guidance toward ventral UNC-6 as well as locally promote presynaptic assembly in the *C. elegans* interneuron AIY (*Hedgecock et al., 1990*; *Colon-Ramos et al., 2007*). In addition, UNC-40 plays critical roles in the formation of dendritic spine-like postsynaptic muscle arms of the body wall muscles in worms (*Dixon and Roy, 2005*; *Alexander et al., 2009*). Intriguingly, this function appears to be independent of UNC-6.

While it is likely that transcription factors ultimately regulate the expression of cell surface molecules to determine the target specificity, few examples are well characterized. In one such example, the even-skipped transcription factor impacts long-range axon guidance choices through regulating a Netrin receptor, UNC-5 (*Labrador et al., 2005*). However, it is largely unknown how cell fate decisions affect local synaptic development and target choices.

One of the conserved developmental pathways to generate cellular diversity is through the lateral signaling system involving the Notch receptor and its ligands. Through contact-dependent, reciprocal feedback loops, Notch and its ligand Delta can generate different cell fates among identical neighboring cells (*Louvi and Artavanis-Tsakonas, 2006*; *Greenwald, 2012*). In *C. elegans*, *lin-12* encodes one of the two homologs of Notch receptor (*Greenwald et al., 1983*; *Greenwald, 1985*; *Wharton et al., 1985*). Extensive literature showed that *lin-12* is required for at least two cell fate decisions: the AC/VU decision and the vulval precursor cell (VPC) specification (*Greenwald, 2005*). In both cases, *lin-12* and its ligands, including *lag-2*, *apx-1* and *dsl-1*, are required to specify alternative cell fates (*Greenwald et al., 1983*; *Seydoux and Greenwald, 1989*; *Wilkinson et al., 1994*; *Shaye and Greenwald, 2002*; *Chen and Greenwald, 2004*). Notch signaling has also been implicated in the development of

the nervous system. For example, the mammalian Notch 1 protein is asymmetrically inherited by one daughter cell of an active cortical progenitor during mitosis, which inhibits the neural differentiation but allows this daughter to retain the progenitor fate (*Ishibashi et al., 1994*; *Chenn and McConnell, 1995*).

The accumulation of nuclear Notch in many post-mitotic neurons is also striking (*Ahmad et al., 1995*; *Sestan et al., 1999*; *Redmond et al., 2000*), prompting the speculation that Notch signaling might also be important for terminally differentiated neurons. Besides functioning in mature nervous system to regulate synaptic specificity in both invertebrates and vertebrates (*de Bivort et al., 2009*; *Alberi et al., 2011*), Notch has also been found to act in the post-mitotic neurons in multiple developmental contexts. Both the canonical Notch pathway and a cytosolic pathway have been shown to play important roles in axonal guidance (*Giniger, 1998*; *Endo et al., 2007*; *Song and Giniger, 2011*). In the cytoplasmic pathway, Notch functions by regulating the activity of abl, enable and Rac GTPases (*Giniger, 1998*; *Song et al., 2010*; *Song and Giniger, 2011*). Studies in *Drosophila* indicate that activation of Notch signaling lead to promotion or sometimes inhibition of axonal growth. In cultured primary *Drosophila* neurons, Notch is localized in developing axons and growth cones and interacts with an axonal abl tyrosine kinase to promote axon extension (*Giniger, 1998*). An opposite example is the dorsal cluster neurons (DCN) of the *Drosophila* brain (*Hassan et al., 2000*). Reduction of Notch activity results in overbranching of DCN axons. Interestingly, the overbranching phenotype cannot be rescued by expressing wild-type Notch in DCN, indicating its non-autonomous requirement. Similarly, dendritic branching is also sensitive to the level of Notch signaling. Several observations indicate a Notch-mediated, contact-dependent inhibition of dendritic branching (*Berezovska et al., 1999*; *Sestan et al., 1999*; *Redmond et al., 2000*). In mammalian brain, significant nuclear localization of Notch 1 was observed in differentiated neurons. Perturbation of Notch 1 activity leads to increased dendritic length and decreased number of branches (*Redmond et al., 2000*). Other in vitro studies further indicated that the intercellular receptor–ligand interaction and the canonical Notch signaling pathway are required for the Notch-dependent dendritic branching inhibition (*Berezovska et al., 1999*; *Sestan et al., 1999*). It is generally assumed that the canonical pathway regulates post-mitotic phenotypes by controlling the expression of guidance molecules. However, the molecular targets of this pathway have not been identified.

With the well-described cell lineage and the stereotyped connectivity between neurons, the nervous system of the nematode *C. elegans* provides an opportunity to study the molecular links between cell fate specification and relevant synaptic target selection. In the egg-laying circuit, the presynaptic neurons, HSNs and VC4/5, form synapses exclusively with type 2 vulval muscles (vm2) but not type 1 vulval muscles (vm1) (*White et al., 1986*) (*Figure 1A*). These synapses are not formed on the cell bodies of the muscles but instead onto the dendritic spine-like postsynaptic membrane protrusions called muscle arms (*White et al., 1986*; *Leung et al., 1999*; *Collins and Koelle, 2012*). The HSN presynaptic development occurs before vm2 muscle arm formation and is independent of vulval muscles. Instead, surrounding guidepost cells instruct local HSN presynapse formation with heterophilic interaction of SYG-1 and SYG-2 (*Shen and Bargmann, 2003*; *Shen et al., 2004*). SYG-2 is expressed in the primary epithelial cells and recruits HSN-expressed SYG-1 to the future presynaptic domain. The development of postsynaptic specializations on the vulval muscles has not been studied. Both the vm1 and vm2 muscle cells are generated from a pair of sex myoblasts (*Foehr and Liu, 2008*). While the lineage, cell morphology and positions of both muscle types are very similar, only vm2 cells, not the vm1 cells, generate muscle arms and serve as the direct postsynaptic target of the HSN and VC neurons. vm1 is connected to vm2 through gap junctions (*White et al., 1986*).

To understand how cell specification was coordinated with synaptic target selection, we explored the cellular and molecular mechanisms underlying vm2 postsynaptic muscle arm development. We found that postsynaptic muscle arm development in egg-laying circuit is mediated by LIN-12/Notch-dependent cell specification. The canonical LIN-12/Notch signaling pathway is specifically activated in the target vm2 cells by the ligand APX-1/DSL expressed in two adjacent cell types, the secondary vulval epithelial cells and the non-target vm1 cells. *lin-12* functions cell-autonomously to up-regulate the activity of the guidance molecules UNC-40/DCC and MADD-2, which in turn generates and guides the development of vm2 postsynaptic muscle arms. The ectopic expression of UNC-40 in vm1 cells that normally do not form muscle arms is sufficient to induce the extension of muscle arm-like membrane structures from vm1, suggesting a deterministic role of LIN-12/Notch and UNC-40/DCC in postsynaptic target selection.

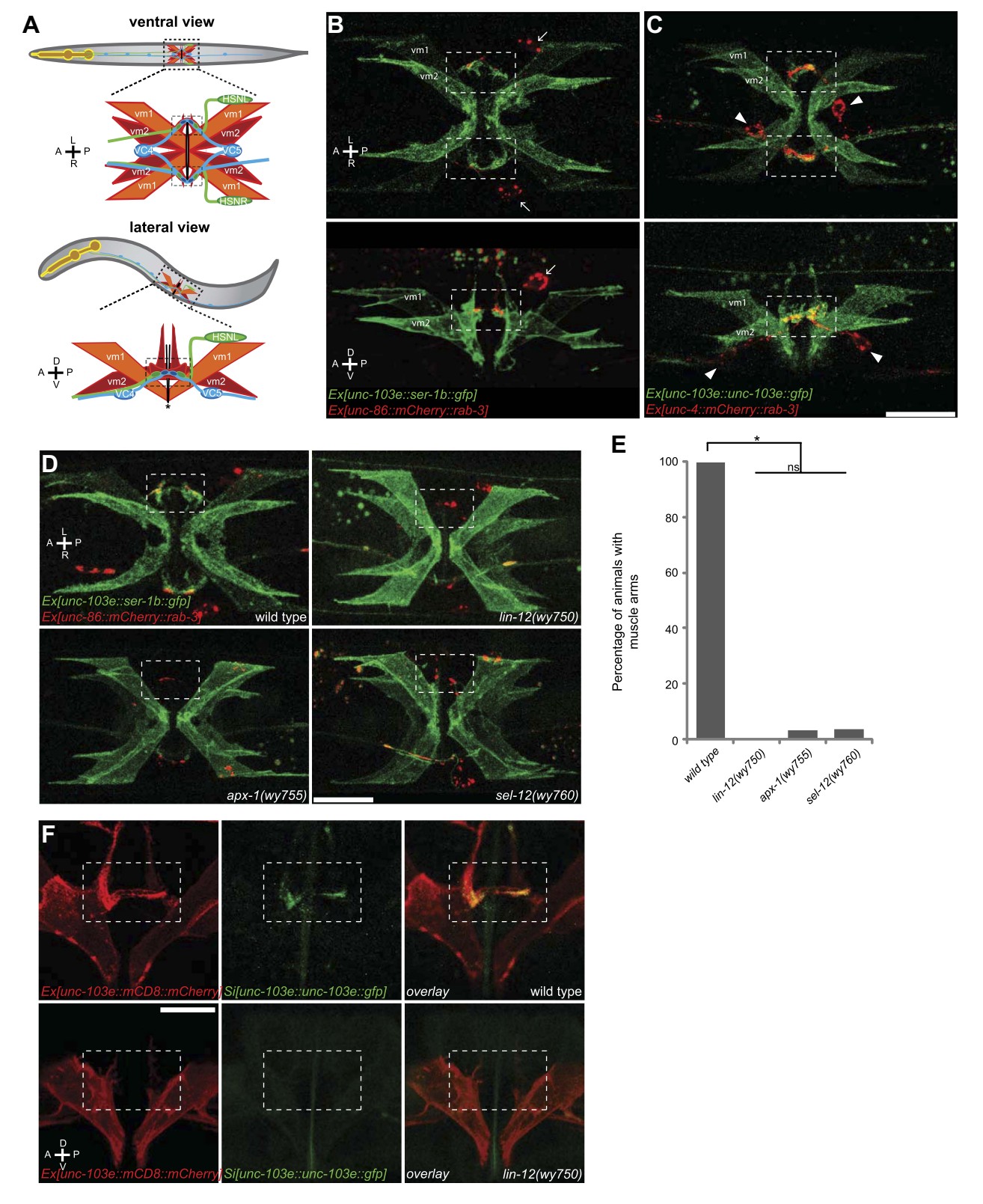

**Figure 1**. The muscle arms of type 2 vulval muscles (vm2) are missing in *lin-12(wy750)*, *apx-1(wy755)* and *sel-12(wy760)* mutants. (**A**) An illustration showing ventral (top) and lateral (bottom) views of *C. elegans* egg-laying circuit. vm1 (orange) and vm2 (red) are vulval muscles. HSN (green) and VC4/5 (blue) are presynaptic motoneurons. Each of the four vm2 cells extends a dendritic spine-like muscle arm laterally, on which it receives synapses with
*Figure 1. Continued on next page*

*Figure 1. Continued*

HSN and VC neurons (boxed areas). Top row of (**A**–**C**) are ventral views; bottom row of (**A**–**C**) are left lateral views. Anterior (A), posterior (P), left (L), right (R), dorsal (D), ventral (V). (**B**) Ventral (top) and lateral (bottom) views of vulval muscles labeled by SER-1B::GFP transgene in a young adult. HSN neurons are labeled by a mCherry::RAB-3 transgene. The boxed areas indicate synaptic regions. Arrows indicate the HSN cell bodies. Note that the laterally extended muscle arms on vm2 cells are colocalized with the HSN presynaptic specializations. (**C**) Ventral (top) and lateral (bottom) views of vulval muscles labeled by a UNC-103E::GFP transgene and VC4/5 neurons labeled by a mCherry::RAB-3 transgene. The boxed areas indicate synaptic regions. Arrowheads indicate cell bodies of VC4 and VC5. Note that the laterally extended muscle arms on vm2 cells are colocalized with the VC4/5 presynaptic specializations. Scale bar is 20 µm. (**D**) Ventral views of the vulval muscles and presynaptic regions of HSN neurons in wild-type, *lin-12(wy750)*, *apx-1(wy755)* and *sel-12(wy760)* animals. The boxed areas indicate synaptic regions. Note the absence of vm2 muscle arms in mutants. Scale bar is 20 µm. (**E**) Quantification of the vm2 muscle arm defects in wild-type, *lin-12(wy750)*, *apx-1(wy755)* and *sel-12(wy760)* animals. * p<0.0001, n.s. no significant difference, chi-squared test, n = 90–114 animals. (**F**) High magnification lateral views of the synaptic region in wild type and *lin-12(wy750)*. vm2 muscle arms are visualized by a single-copy UNC-103E::GFP transgene. Vulval muscle morphology is labeled by mCD8::mCherry. The boxed areas indicate synaptic regions. Scale bar is 10 µm.

The following figure supplements are available for figure 1:

**Figure supplement 1**. vm2 muscle arms are the postsynaptic specializations.

**Figure supplement 2**. vm2 muscle arm development is a guidance event independent of presynaptic specialization.

**Figure supplement 3**. Schematic model of LIN-12/Notch signaling pathway.

**Figure supplement 4**. *lin-12(wy750)* has specific vm2 muscle arm defects.

**Figure supplement 5**. Schematic demonstrations of the quantitative measurements of vulval muscle morphology.

**Figure supplement 6**. Quantifications of the vulval muscle morphology.

## Results

### Visualization of postsynaptic muscle arms of type 2 vulval muscles

The *C. elegans* egg-laying circuit consists four type 1 vulval muscle cells (vm1) and four type 2 vulval muscle cells (vm2), which form two X-shaped structures near the vulval opening (*Figure 1A*). While vm1 and vm2 are from a common lineage, are physically adjacent and share many molecular components, the egg-laying motoneurons HSN and VC4/5 form serotoninergic and cholinergic chemical synapses exclusively on vm2 (*White et al., 1986*; *Waggoner et al., 1998*; *Duerr et al., 2001*; *Kim et al., 2001*). vm1 is mainly controlled by vm2 via gap junctions (*White et al., 1986*). Besides forming major synaptic connection on HSN and VC4/5, vm2 also receive some additional synapses with the ventral nerve cord (Personal communication with Michael Koelle, November 2012). To understand how vm2 is differentiated as the postsynaptic target and specifies the synaptic specializations, we first developed tools to visualize the egg-laying synapses. We fused presynaptic protein RAB-3 with mCherry (*mCherry::rab-3*) to label presynaptic vesicles in HSN or VC neurons, receptively (*Baumeister et al., 1996*; *Lickteig et al., 2001*; *Shen and Bargmann, 2003*). On the postsynaptic side, we expressed *ser-1b::gfp* or *unc-103e::gfp*, a GFP tagged serotonin receptor or a voltage-gated potassium channel, in both vm1 and vm2 to visualize the postsynaptic specializations as well as the muscle cell morphology (*Figure 1B,C*). Consistent with EM reconstruction, we found that the HSN presynaptic specializations were closely juxtaposed against the dendritic spine-like membrane protrusions (muscle arms) from the vm2 cell bodies when viewed from both the lateral and ventral sides (*Figure 1B*). The muscle arms always extend from stereotyped subcellular locations, medially towards the center of the vulva. The muscle arms from the anterior and posterior vm2 cells often contact each other and form a 'bridge-like' structure that is best visualized from the ventral view (*Figure 1B,C*, top row). Similar apposition was also observed between VC4/5 presynaptic specializations and the vm2 muscle arms (*Figure 1C*). While high level of UNC-103E::GFP expression labels both the muscle arms and the vm cell bodies (*Figure 1C*), low level of transgenic expression of UNC-103E::GFP with an integrated single-copy transgene showed that UNC-103E was dramatically enriched on vm2 muscle arms

(*Figure 1—figure supplement 1*, middle), indicating that the muscle arms represent functional postsynaptic specializations (*Collins and Koelle, 2012*).

## vm2 postsynaptic development requires vulval epithelial cells

Because most egg-laying synapses are formed onto the vm2 muscle arms, we investigated the cellular and molecular mechanisms mediating the vm2 muscle arm formation to gain insight into the postsynaptic development and synaptic target selection. Developmental studies showed that HSN axon guidance and presynaptic formation preceded both axon growth of VC neurons and vulval muscle differentiation (data not shown). We therefore asked whether the HSN neurons are required for vm2 specification and muscle arm development by examining the vm2 morphology in the *egl-1(n986)* mutant, in which the HSN neurons undergo programmed cell death in embryonic stage (*Conradt and Horvitz, 1998*). Surprisingly, in all of the examined animals, the morphology and position of vm2 muscle arms were indistinguishable from the wild type counterparts (*Figure 1—figure supplement 2A*). Several results further suggested that neither HSN nor VC neurons contributed to the vm2 muscle arm position. First, in a portion of the *egl-1(n986);lin-39(n709)* double mutant animals which lacked both HSN and VC neurons (*Clark et al., 1993*), the vm2 muscle arms still existed (*Figure 1—figure supplement 2B*). Second, in *unc-104(e1265)* mutant animals, in which the synaptic vesicles were absent from the presynaptic regions in both HSN and VC neurons (*Shen and Bargmann, 2003*), vm2 muscle arm development was not affected (*Figure 1—figure supplement 2A*). These results indicate that presynaptic components are dispensable for vm2 postsynaptic development.

If vm2 postsynaptic differentiation and muscle arm development are independent of presynaptic components, we asked whether the surrounding vulval epithelial cells that physically contact vm2 provide guidance for vm2 specification and muscle arm development. We examined the vm2 muscle arm morphology in *lin-3(e1417)* mutant animals, in which vulval epithelial cell fate is abnormal (*Hill and Sternberg, 1992*), and observed severe defects on both vulval muscle morphology and muscle arm development (*Figure 1—figure supplement 2A*). This result suggests that the vulval epithelial cells are required for normal vulval muscle differentiation and morphogenesis, and therefore might be also important for vm2 postsynaptic muscle arm development.

## A mutation in *lin-12* affects the vm2 postsynaptic development

To understand the molecular mechanism of vm2 postsynaptic muscle arm formation, we performed a forward genetic screen on the transgenic strain labeling both HSN presynaptic specializations and vm2 postsynaptic muscle arms. We searched for mutants with normal vm2 differentiation and normal vulval morphology but with abnormal vm2 muscle arms. In total, six individual mutants representing three complementation groups were isolated in which the vm2 muscle arms were largely absent (*Figure 1D,E* and *Table 1*). Mapping and molecular identification of these mutations showed that they affected genes in LIN-12/Notch signaling pathway including *lin-12/Notch*, *apx-1/DSL* and *sel-12/presinilin* (*Table 1* and *Figure 1—figure supplement 3*).

In *lin-12(wy750)* mutant animals, while the muscle morphology of both vm1 and vm2 were largely normal, the SER-1B::GFP labeled vm2 muscle arm structures were completely absent (*Figure 1D,E*). Furthermore, the single-copy insertion of the transgene UNC-103E::GFP potassium channel which specifically labels the vm2 muscle arms, as well as the artificial membrane marker mCD8::mCherry that delineates the entire outline of vulval muscles, was also completely absent in the muscle arm areas, suggestive of a complete loss of the muscle arm structure (*Figure 1F*). Both HSN and VC presynaptic markers appeared normal in the *lin-12(wy750)* mutant, indicating that the mutation specifically affected postsynaptic differentiation and development (*Figure 1—figure supplement 4B,D*). Surprisingly, we found no vulval morphogenesis defect in *lin-12(wy750)* mutants either by differential interference contrast (DIC) microscopy or the localization pattern of the *ajm-1::gfp* apical junction marker at several development stages (*Figure 1—figure supplement 4A* and data not shown) (*Koppen et al., 2001*). Two other mutants isolated from the same screen, *apx-1(wy750)* and *sel-12(wy760)*, were very similar to the *lin-12(wy750)* mutant in their specific defective vm2 muscle arm development and normal vulval muscle development (*Figure 1D* and data not shown).

Vulval muscles, together with the uterine muscles, are produced through three rounds of cell divisions from a single sex myoblast cell, which is derived from the postembryonic mesodermal lineage (M lineage). We considered the possibility that the absence of muscle arms in vm2 could be due to the change of vm2 cell fate to a different cell. We therefore examined several cell-type-specific markers

**Table 1.** Mutants isolated from screen for genes required for vm2 postsynaptic target selection

| Complementation group | Allele | Mutation | Homolog |
|---|---|---|---|
| lin-12 | wy750 | G473R(g1417a) | Notch |
| apx-1 | wy754 | G208E(g623a) | Dsl |
| | wy755 | G159E(g476a) | |
| | wy766 | C217Y(g650a) | |
| sel-12 | wy756 | W184stop(g552a) | Presenilin |
| | wy760 | G373D(g1118a) | |

for each differentiation stage: the *hlh-8::gfp* that is specifically expressed in undifferentiated cells of M lineage (*Harfe et al., 1998b*) and an *egl-15::gfp* reporter which is exclusively expressed in vm1 cells in egg-laying system (*Harfe et al., 1998a*; *Huang and Stern, 2004*). We also examined *rgs-2::gfp* that labeled uterine muscles (*Dong et al., 2000*). In *lin-12(wy750)* mutants, all three markers exhibits wild type pattern, suggesting that many molecular signatures of the M lineage are intact in the *lin-12(wy750)* mutants, while the development of the postsynaptic specializations completely fails (*Figure 1—figure supplement 4C,E,F*). Furthermore, since both vm2 and vm1 adopt complex cell morphology and stereotyped positions, we reasoned that the morphology and position of these muscles must reflect the execution of cell-specific developmental programs. We therefore quantitatively analyzed the morphology and relative position of vm1 and vm2 (*Figure 1—figure supplement 5*). In both wild type and *lin-12(wy750)* mutant, the position where vm1 is linked to the body wall is always on the dorsal side of the connection between vm2 and body wall. No significant difference was found on the intersection angle between the vulval muscle and the vulval slit, as well as the relative size of vm2 to vm1 ($S_{vm2}/S_{vm1}$) (*Figure 1—figure supplement 6*). In summary, all these observations supports the notion that the muscle defects in *lin-12* mutants appear to be restricted to the postsynaptic development in the vm2 cells.

## Asynchronous vulval muscle Ca²⁺ transients in vm2 muscle arm mutants

If vm2 muscle arms play crucial functional roles in the synaptic connections of the egg-laying circuit, mutants with defective vm2 muscle arm development should show altered egg-laying behavior. As a measure of this behavior, we examined the eggs freshly-laid by the wild-type animals and the muscle arm defective mutants. First, the muscle arm defective mutants laid notably fewer eggs compared to the wild-type animals (*Figure 2A*). A more direct and sensitive way to analyze egg-laying is to quantify the developmental stages of the freshly-laid eggs. Indeed, we found that these mutants laid more late-stage eggs and fewer early-stage eggs compared with wild-type animals (*Figure 2B*), indicative of defects in the egg-laying circuit.

We then investigated the functional consequences of the vm2 muscle arm defects seen in *lin-12(wy750)* and *apx-1(wy755)* mutants, and the physiological basis for their reduced egg-laying. By co-expressing the Ca²⁺ reporter GCaMP3 and the calcium-insensitive fluorescent protein mCherry in the vulval muscles, we were able to perform ratiometric Ca²⁺ imaging in behaving wild-type, *lin-12(wy750)*, and *apx-1(wy755)* mutant animals. In wild-type animals, simultaneous Ca²⁺ transients in the anterior and posterior vulval muscles, which we call 'double' transients, drive synchronous muscle contraction and release of eggs (*Collins and Koelle, 2012*) (*Figure 2C*). Smaller Ca²⁺ transients (magnitude < 150% ΔR/R) drive small vulval muscle twitching contractions that do not result in release of eggs. Such twitch transients are sometimes 'single' transients, limited to either the anterior or posterior muscles, or smaller magnitude double transients. Double transients result in larger peaks in our ratiometric trace recordings because the anterior and posterior signals are summed (*Figure 2E* and *Video 1*).

Despite their lack of postsynaptic muscle arms, we found that *lin-12(wy750)* and *apx-1(wy755)* vm2 muscle arm mutants were still able to execute single anterior and posterior vulval muscle Ca²⁺ transients, which was presumably due to the intact minor synaptic inputs from the ventral nerve cord alongside the vulval muscles (*Collins and Koelle, 2012*). However, double Ca²⁺ transients were often asynchronous, with significant delays between anterior and posterior vulval muscle transients that resulted in uncoordinated vulval muscle contractions that were not able to release eggs (*Figure 2D* and *Video 2*).

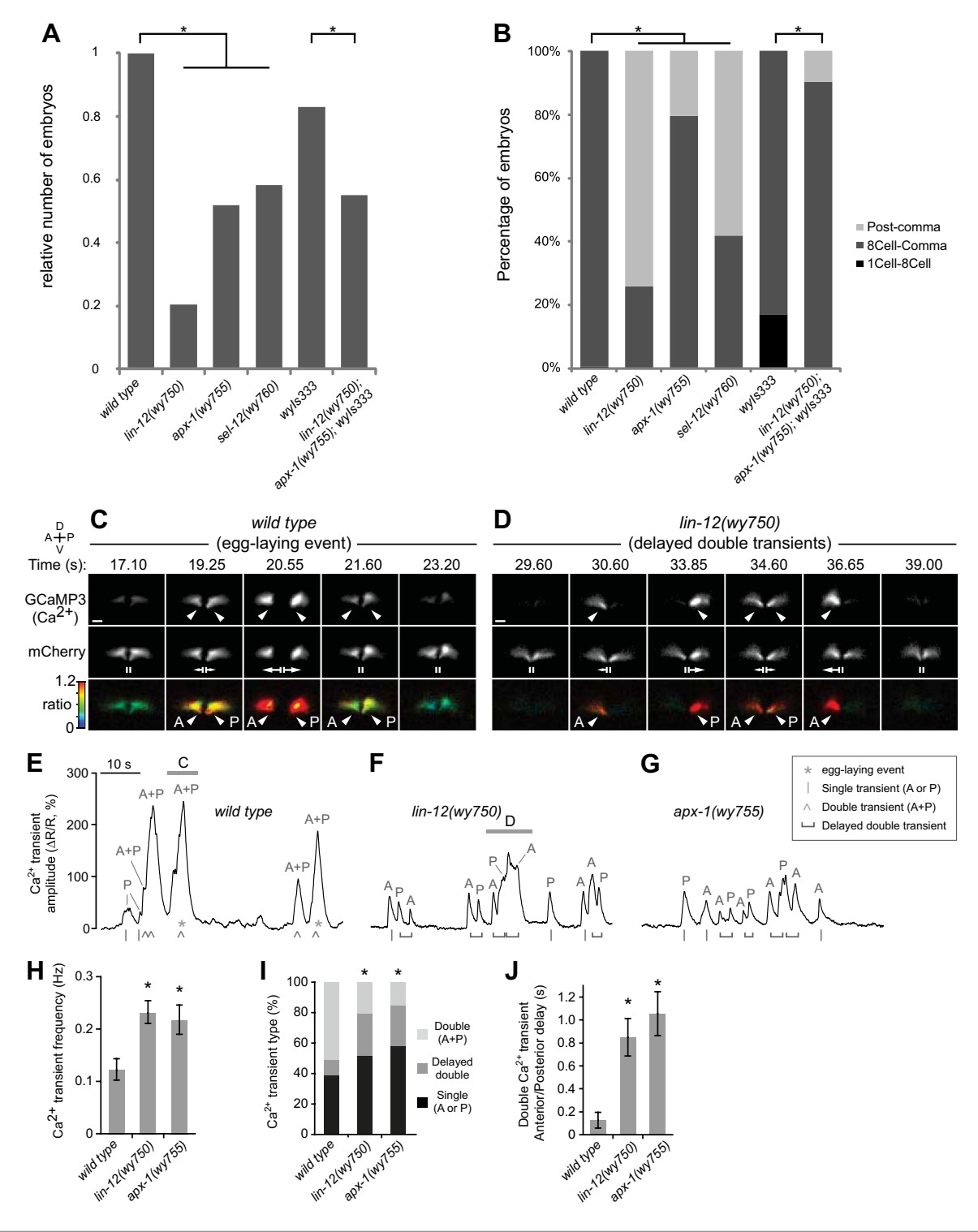

**Figure 2**. Muscle arm defective mutants have egg-laying behavioral and physiological defects. (**A**) Relative numbers of embryos freshly-laid by wild-type and mutant animals. *wyIs333* is the marker that double-labels the HSN presynaptic specializations and vulval muscles. *p<0.0001, Fisher's exact test, n = 62–303 embryos. (**B**) Percentage of different stages of embryos freshly-laid by wild-type and mutants animals. *p<0.0001, Fisher's exact test, n = 62–303 embryos. (**C**) and (**D**) Ratiometric Ca2+ imaging in the vulval muscles in behaving wild-type (**C**) and *lin-12(wy750)* (**D**) animals. GCaMP3 (top) and mCherry (middle) were co-expressed in the vulval muscles and the GCaMP3/mCherry fluorescence ratio (bottom) was used to record Ca2+ transients (arrowheads). Time points are shown from ***Video 1*** (Wild type) and ***Video 2*** (*lin-12(wy750)* mutant). Vertical lines (II) indicate the vulval muscles at rest, and arrows indicate vulval muscle twitches (small) and egg-laying (large) contractions. Anterior (A), posterior (P), left (L), right (R), dorsal (D), ventral (V). Scale bars are 10 μm. *Figure 2. Continued on next page*

Figure 2. Continued

(**E–G**) traces of vulval muscle GCaMP3/mCherry ratio change (ΔR/R) from the same wild-type (**E**) and *lin-12(wy750)* mutant (**F**) animals shown above (horizontal bars) and in the *apx-1(wy755)* mutant (**G**). Also indicated are egg-laying events (\*), single transients limited to the anterior or posterior (A or P) vulval muscles (|), double transients occurring simultaneously in both anterior and posterior (A + P) vulval muscles (^), or delayed double transient where anterior and posterior transients within a body bend are separated by a visually discernible interval (horizontal bracket). (**H–J**) Quantitations of vulval muscle $Ca^{2+}$ signaling (6-min recording per animal, 11 or 12 animals per genotype; error bars indicate 95% confidence intervals). (**H**) *lin-12(wy750)* and *apx-1(wy755)* mutants have more frequent $Ca^{2+}$ transients than does the wild type. \*p<0.0001, one-way ANOVA, n = 215–488. (**I**) Fewer synchronous double (A + P) $Ca^{2+}$ transients in *lin-12(wy750)* and *apx-1(wy755)* mutants. \*p<0.0001, chi-squared test, n = 200–305. (**J**) Consecutive anterior and posterior $Ca^{2+}$ transients are delayed in *lin-12(wy750)* and *apx-1(wy755)* mutants. \*p<0.0001; one-way ANOVA, n = 123–190.

In the ratiometric traces, these 'delayed double' $Ca^{2+}$ transients could be resolved into two or more peaks instead of one large peak as in the wild type (*Figure 2E–G*). As a result, *lin-12(wy750)* and *apx-1(wy755)* mutants had a twofold increase in the frequency of $Ca^{2+}$ transient peaks (*Figure 2H*). We also found that *lin-12(wy750)* and *apx-1(wy755)* mutants had significantly fewer synchronous double transients compared to the wild type (*Figure 2I*). In the rare instances when strong single transients or asynchronously initiated anterior/posterior $Ca^{2+}$ transients of *lin-12(wy750)* and *apx-1(wy755)* mutants led to opening of the vulva, egg-laying would still occur, indicating that their egg-laying defect was not caused by structural problems in the vulva itself (*Video 3*).

To quantify the anterior/posterior coordination in double $Ca^{2+}$ transients, we determined the delay between consecutive anterior and posterior $Ca^{2+}$ transient peaks that occurred during a single locomotor body bend. In wild-type animals, the average anterior/posterior delay was only ~100 ms (*Figure 2J*), as nearly all double transients were not resolved into separate anterior/posterior peaks at the temporal and spatial resolution of our ΔR/R traces (*Figure 2E*). In both *lin-12(wy750)* and *apx-1(wy755)* mutants, the average anterior/posterior delay in double $Ca^{2+}$ transients increased to ~800 ms (*Figure 2J*) and anterior/posterior transients were easily resolved (*Figure 2F,G*). Together, these results show that the vm2 muscle arms lacking in *lin-12(wy750)* and *apx-1(wy755)* mutants are not necessary for excitation of the vulval muscles, but suggest they are required for the synchronous excitation of anterior and posterior vulval muscles needed to successfully execute egg-laying behavior.

## Muscle arm development requires a high level of LIN-12/Notch signaling

LIN-12/Notch pathway was originally discovered through its effects on anchor cell and vulval precursor cell fates (*Greenwald et al., 1983*; *Sternberg, 1988*; *Seydoux and Greenwald, 1989*). It was surprising that we isolated multiple alleles of genes in the LIN-12/Notch pathway, which did not show obvious phenotypes in vulval morphogenesis, except for one *sel-12* early-stop allele *wy756* (*Table 1*). To understand how vm2 muscle arm development was specifically affected, we identified the molecular lesions of these mutant alleles.

Sequence analyses revealed that *lin-12(wy750)* contained a single missense mutation (G473R) in one of the conserved extracellular EGF domains of the LIN-12 protein (*Table 1*). Two additional results indicated that *lin-12(wy750)* was a loss-of-function allele. First, trans-heterozygotes between *lin-12(wy750)* and a canonical loss-of-function allele *lin-12(n676n930)* presented similar muscle arm defects (data not shown). Second, a *lin-12* genomic fragment including 5 kb of upstream

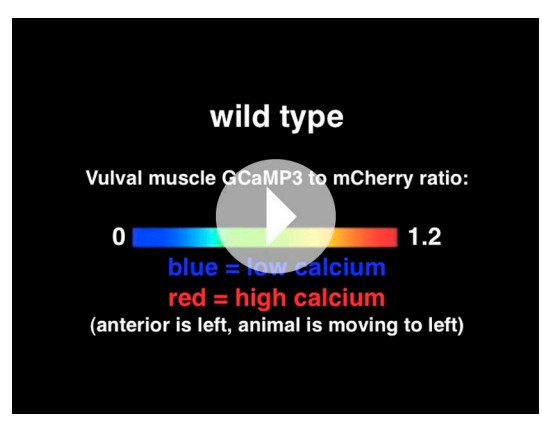

**Video 1.** Synchronous $Ca^{2+}$ transients in the vulval muscles of wild-type animals drive egg-laying behavior. Intensity-modulated ratiometric imaging of wild-type *C. elegans* expressing the $Ca^{2+}$ sensor GCaMP3 and soluble mCherry in the vulval muscles from the *unc-103e* promoter at 20 fps. Change in the GCaMP3 to mCherry fluorescence ratio is indicated by a rainbow scale from 0 (dark blue) to 1.2 (red). Egg-laying events are observed at 23 and 56 s. Still images from this video are shown in *Figure 2C*, and the traces of ΔR/R and vulval muscle area are shown in *Figure 2E*.

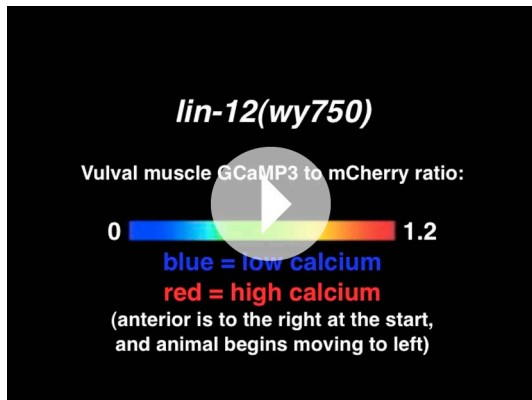

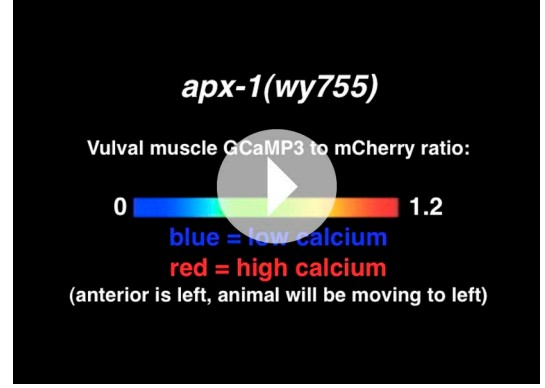

**Video 2** *lin-12* mutants have asynchronous vulval muscle Ca²⁺ transients leading to uncoordinated contractions. Intensity-modulated ratiometric imaging of *lin-12(wy750)* mutants expressing the Ca²⁺ sensor GCaMP3 and soluble mCherry in the vulval muscles from the *unc-103e* promoter at 20 fps. Change in the GCaMP3 to mCherry fluorescence ratio is indicated by a rainbow scale from 0 (dark blue) to 1.2 (red). Still images from this video are shown in **Figure 2D**, and the traces of ΔR/R and vulval muscle area are shown in **Figure 2F**.

**Video 3** Asynchronous vulval muscle Ca²⁺ transients in *lin-12* and *apx-1* mutants can still lead to uncoordinated vulval opening and egg laying. Intensity-modulated ratiometric imaging of *apx-1(wy755)* mutants expressing the Ca²⁺ sensor GCaMP3 and soluble mCherry in the vulval muscles from the *unc-103e* promoter at 20 fps. Change in the GCaMP3 to mCherry fluorescence ratio is indicated by a rainbow scale from 0 (dark blue) to 1.2 (red). At 6 and 18 s, there was sufficient vulval opening to permit an egg-laying event.

regulatory sequence and the entire coding region of *lin-12* strongly rescued the muscle arm phenotype of *lin-12(wy750)* mutants (**Figure 4A**). Since strong loss-of-function alleles of *lin-12*, *apx-1* and *sel-12* cause lethality, dramatic vulval morphogenesis or general muscle morphology phenotypes, which are all absent in *lin-12(wy750)* mutant animals, it is likely that the missense mutation we isolated represents a partial loss-of-function allele. Three *apx-1* mutant alleles independently identified from the screen also contain missense mutations in the extracellular domain. One early-stop mutation in the *sel-12* coding region leads to additional pleiotropic effects on both vulval muscle patterning and vulval morphogenesis, while the other missense mutation of *sel-12* causes specific muscle arm defect (**Table 1**). Together, these data suggest that partial loss of LIN-12/Notch signaling activity spares vulval morphogenesis but specifically impairs vm2 specification and muscle arm development, and that muscle arm development might require a higher level of LIN-12/Notch signaling activity compared with earlier developmental events (see 'Discussion').

## LIN-12 functions at the L4 stage to specify vm2 postsynaptic identity and muscle arm development

To further distinguish whether LIN-12/Notch signaling is directly involved in the vm2 muscle arm development or indirectly affects vm2 development due to its role in vulval morphogenesis, we investigated the spatial and temporal requirements of *lin-12*.

Previous studies suggest a late requirement for *lin-12* activity to allow proper egg-laying behavior (**Sundaram and Greenwald, 1993**). The 'late egg-laying defect' appears to be independent of AC/VU decision and vulval precursor cell (VPC) specification regulated by LIN-12/Notch signaling (**Sundaram and Greenwald, 1993**). We hypothesized that the late egg-laying defect could be caused by the defects of vm2 muscle arm development. In order to test this idea, we examined a temperature sensitive allele, *lin-12(n676n930*ts*)*, in which LIN-12 activity is normal at the permissive temperature (15°C) but disrupted at the restrictive temperature (25°C). Consistent with previous behavioral studies, *lin-12(n676n930)* mutant animals grown at the permissive temperature had no egg-laying defect and showed intact vulval muscle morphology and vm2 muscle arms, whereas animals grown at the restrictive temperature retained significantly more unlaid eggs inside their bodies and displayed vulval morphogenesis and vulval muscle arm defects. We then shifted *lin-12(n676n930)* mutant animals from the permissive temperature to the restrictive temperature at different larval stages and examined the vulval muscle morphogenesis and vm2 muscle arm development in the resulting adults. Most of the

animals shifted from 15°C to 25°C prior to mid-L4 stage showed muscle arm defects similar to those of *lin-12(wy750)*, whereas the wild-type muscle arm phenotype was observed in most of *lin-12(n676n930)* animals that were shifted after mid-L4 stage (*Figure 3A*).

In a temperature down-shift experiment, designed to circumvent LIN-12's role in VPC specification, we first kept *lin-12(n676n930)* mutant animals at the permissive temperature until the early-L3 stage, when the AC/VU decision and VPC specification processes were complete but vulval muscle cells had not yet been generated (*Sundaram and Greenwald, 1993*; *Foehr and Liu, 2008*). Early-L3 animals were then transferred to 25°C and shifted back to 15°C at different development stages. In all the animals we examined, no severe vulval morphogenesis or general vulval-muscle patterning defects were observed. More than 80% of the animals still showed defective muscle arms when shifted from 25°C to 15°C at mid-L4 (*Figure 3B*). The upshift and downshift experiments together suggest a novel function of *lin-12* at the L4 stage in regulating the postsynaptic muscle arm development and synaptic target choice. This *lin-12* function is independent of other known roles of *lin-12*, all of which occur prior to early-L4 stage.

## LIN-12/Notch signaling functions cell-autonomously in vm2 to regulate postsynaptic specification and muscle arm development

We next investigated where *lin-12* functions to regulate the vm2 muscle arm development by analyzing the expression of *lin-12*. A type II short trans-splicing leader sequence followed by mCherry (*SL2::mCherry*) was inserted into a fosmid containing the *lin-12* genomic sequence at the 3' end of the *lin-12* coding region (*lin-12::SL2::mCherry*). This recombineered fosmid fully rescued the *lin-12* muscle arm phenotype (*Figure 4A* and data not shown), suggesting that this construct should contain all upstream, downstream and intronic regulatory elements crucial for proper expression and function of *lin-12*. Transgenic animals carrying this construct showed mCherry expression in a number of cell types including both the vm2 muscle cells (outlined by dashed lines in *Figure 4A,B*) and the vulval epithelial cells (*Figure 4A,B*). Strikingly, no detectable *lin-12* expression was observed in vm1 muscle cells. To further understand whether *lin-12* is required in the vulval epithelial cells or vm2, we injected this construct into the *lin-12(wy750)* mutant and analyzed mosaic animals in which expression of this transgene was lost in some of the tissues. Forty-five of 48 animals with *lin-12* expressed in the vm2 cells (vm2+) showed a rescued, wild-type muscle arms (*Figure 4A,C*), whereas the muscle arm defect was never rescued in any of the examined animals (>50) lacking *lin-12* expression in the vm2 cell (vm2-) (*Figure 4B,C*). On the contrary, *lin-12* expression in vulval epithelial cells was not correlated with the rescue of the muscle arm defects (data not shown). The mosaic rescue analyses argue that *lin-12* functions cell-autonomously in vm2 to specify vm2 postsynaptic development through regulating muscle arm formation.

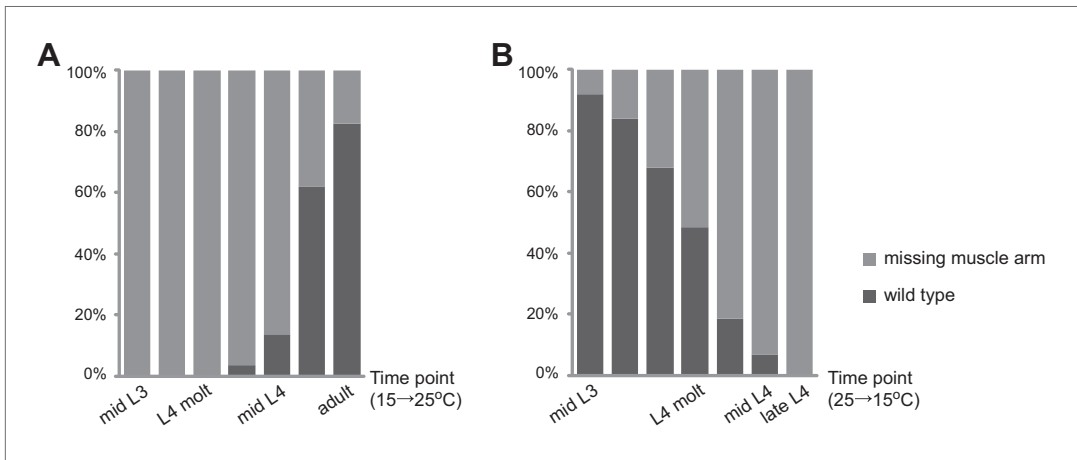

**Figure 3**. Temporal requirement of LIN-12 in vm2 muscle arm formation. (**A**) and (**B**) Temperature up-shift (**A**) and down-shift (**B**) experiments of *lin-12(n676n930)*. The percentages of animals with normal or defective muscle arm phenotype are indicated by dark or light gray bars, respectively. The time points when animals are shifted are shown on X-axis.

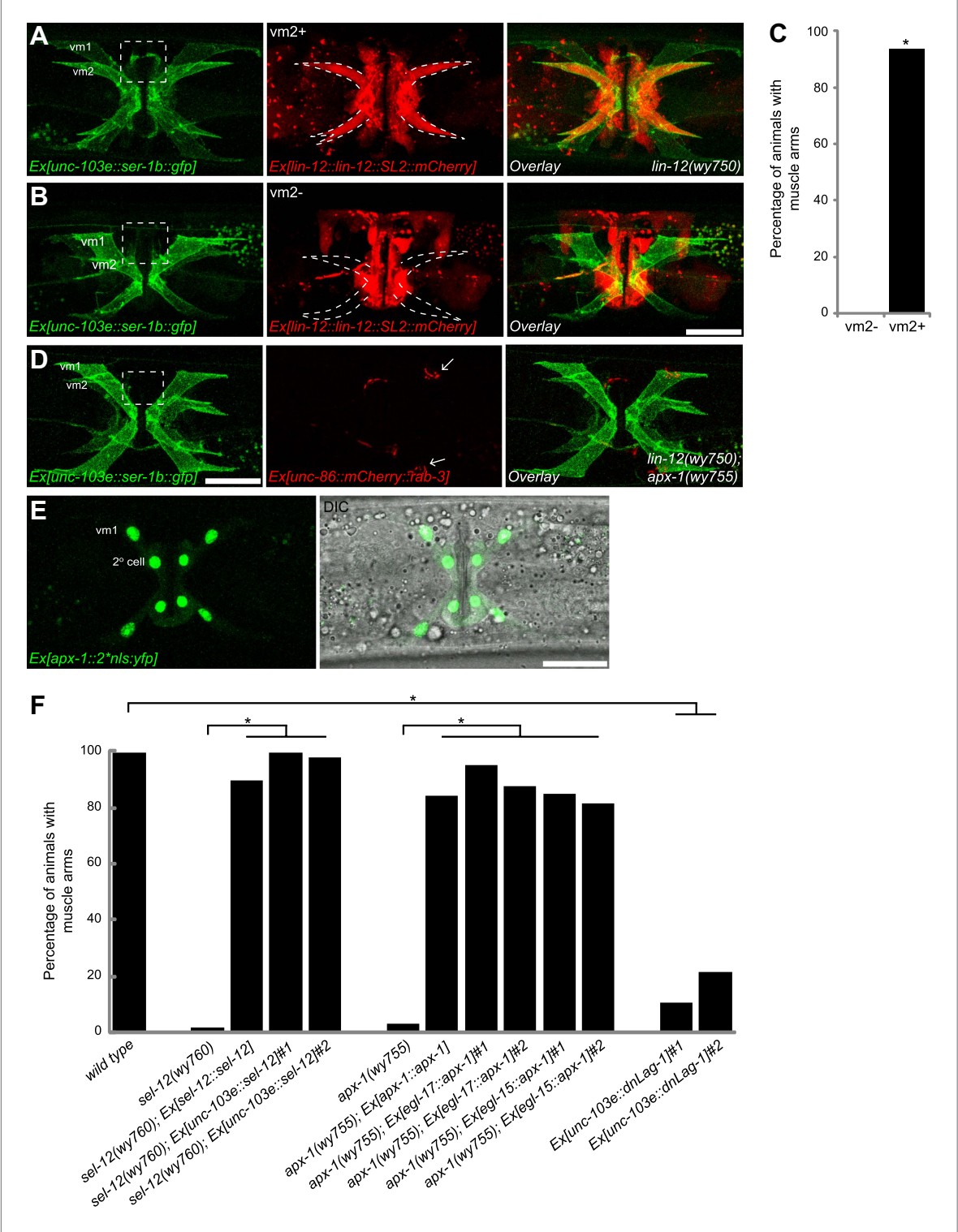

**Figure 4**. Cell-autonomous requirement of LIN-12 in vm2. (**A**) and (**B**) Representative images of *lin-12(wy750)* mutant animals with non-mosaic (**A**) and mosaic (**B**) expression of *lin-12::SL2::mCherry*. Left column shows the vulval muscles labeled by SER-1B::GFP. Middle column shows the *lin-12* expression patterns. Right column shows the overlaid images. Note that the complete *lin-12* expression (vm2+) rescues muscle arm defects of *lin-12(wy750)* (**A**). Lacking of *lin-12* expression in vm2 (vm2-) abolishes its capacity to rescue muscle arm defects (**B**). Boxed areas indicate synaptic regions. Dashed lines indicate the morphology of vm2. Scale bar is 20 μm. (**C**) Quantification of the vm2 muscle arm phenotypes in the mosaic transgenic animals. *p<0.0001, Fisher's exact test, n = 48–50. (**D**) Vulval muscle morphology in the *apx-1(wy755); lin-12(w750)* mutant animals.
*Figure 4. Continued on next page*

*Figure 4. Continued*

Boxed area indicates synaptic region. Arrows indicate HSN cell bodies. Scale bar is 20 μm. (**E**) Epifluorescence and DIC images showing the *apx-1* expression pattern in young adult. The four cells in the center of the image are 2° vulval epithelial cells. The four cells at the periphery are vm1 cells. Scale bar is 20 μm. (**F**) Cell-autonomous requirement of *apx-1*, *sel-12* and *lag-1*. Quantification of muscle arm phenotypes in animals with different genotypes indicated on the X-axis. *, P<0.0001, Fisher's exact test, n = 40–57.
The following figure supplements are available for figure 4:

**Figure supplement 1**. APX-1 expression pattern and the relative position of vm2 muscle arms.

LIN-12 is a membrane-associated protein that can be activated by binding with its ligand. Based on the cell-autonomous function of *lin-12* in vm2, we made two predictions. First, the ligand(s) of LIN-12 should be localized in the surrounding cells that directly contact vm2 to activate LIN-12/ Notch pathway. Second, downstream components of LIN-12/Notch signaling pathway should also function in vm2 cells to regulate vm2 specification and muscle arm development. APX-1/Dsl is a good candidate as the relevant ligand because *apx-1(wy755)*, but not other known LIN-12 ligand mutants including *dsl-1* and *lag-2*, showed the muscle arm phenotype similar to that of *lin-12(wy750)* (**Figure 1D** and data not shown). A *lin-12(wy750); apx-1(wy755)* double mutant did not show an enhanced muscle arm defect (**Figure 4D**) or more severe egg-laying phenotype (**Figure 2A**), further suggesting that *apx-1* and *lin-12* function in the same pathway. To further understand where *apx-1* functioned, we examined the expression pattern of *apx-1* with an integrated transgenic expression construct (Personal communication with Iva Greenwald, June 2012). We observed the expression of *apx-1* in both secondary vulval epithelial cells and vm1 muscle cells during the L4 stage, when LIN-12/Notch signaling was activated to promote vm2 muscle arm development (**Figure 4E** and **Figure 4— figure supplement 1A,B**). Both secondary vulval epithelial cells and vm1 cells contact vm2 muscle cells from early L4 stage when vulval muscles are generated (**Figure 4—figure supplement 1C**, data not shown, and personal communication with Kelly Liu, July 2012). In fact, we observed that the vm2 muscle arms consistently tracked the cell junctions between the primary and secondary vulval epithelial cells (**Figure 4—figure supplement 1C** and data not shown). To further understand the cellular requirement of *apx-1*, we created two constructs in which the *apx-1* cDNA was cloned after two different promoters. The *egl-15* promoter supports expression specifically in the vm1 cells (**Harfe et al., 1998a**; **Eimer et al., 2002**; **Huang and Stern, 2004**), while the *egl-17* promoter drives expression in the secondary vulval epithelial cells from early-L4 stage (**Shaye and Greenwald, 2002**). Cell-specific expression of *apx-1* in either secondary vulval epithelial cells or vm1 muscle cells rescued the muscle arm defects of *apx-1(wy755)* mutants (**Figure 4F**). These data strongly suggest that *apx-1* functions in the vm1 and secondary vulval epithelial cells to activate LIN-12/Notch signaling in vm2, the postsynaptic target cells.

Our genetic screen also yielded two alleles (*wy760* and *wy756*) of *sel-12,* the *C. elegans* homolog of presenilin (**Table 1** and **Figure 1—figure supplement 3**). *sel-12* encodes the γ-secretase that releases the LIN-12/NOTCH intracellular domain (NICD) by ligand-induced cleavage (**Levitan and Greenwald, 1995**; **Ray et al., 1999a**, **1999b**; **Selkoe and Kopan, 2003**). NICD is then transported into cell nucleus where it serves as a transcription factor together with LAG-1/CSL to promote gene expression (**Christensen et al., 1996**; **Andersson et al., 2011**) (**Figure 1—figure supplement 3**). To understand where *sel-12* was required, we performed cell-specific rescue experiments. We found that expression of *sel-12* in vulval muscle cells completely rescued the vm2 muscle arm phenotype of *sel-12(wy760)*, consistent with the cell-autonomous requirement of LIN-12/Notch signaling in vm2 (**Figure 4F**). Additionally, specific expression of a dominant-negative *lag-1* construct (**Kato et al., 1997**) in the vulval muscles phenocopied the muscle arm defects observed in *lin-12(wy750)*, *apx-1(wy750)* and *sel-12(wy760)* mutants (**Figure 4F**), suggesting that the canonical LIN-12/Notch pathway is likely involved by controlling the transcription of downstream genes. Moreover, the expression of a *hlh-29* transcription reporter (*hlh-29::gfp*), a known target gene of the canonical LIN-12/Notch pathway in vm2, was also dramatically down-regulated in *lin-12(wy750)*, *apx-1(wy755)* and *sel-12(wy760)* mutants (**Figure 5A** and **Figure 5—figure supplement 1A**) (**Fischer and Gessler, 2007**; **McMiller et al., 2007**). Together, these results strongly support the model that APX-1 expressed in the secondary vulval epithelial cells and vm1 cells activates the canonical LIN-12/Notch pathway in vm2 to promote the formation of the

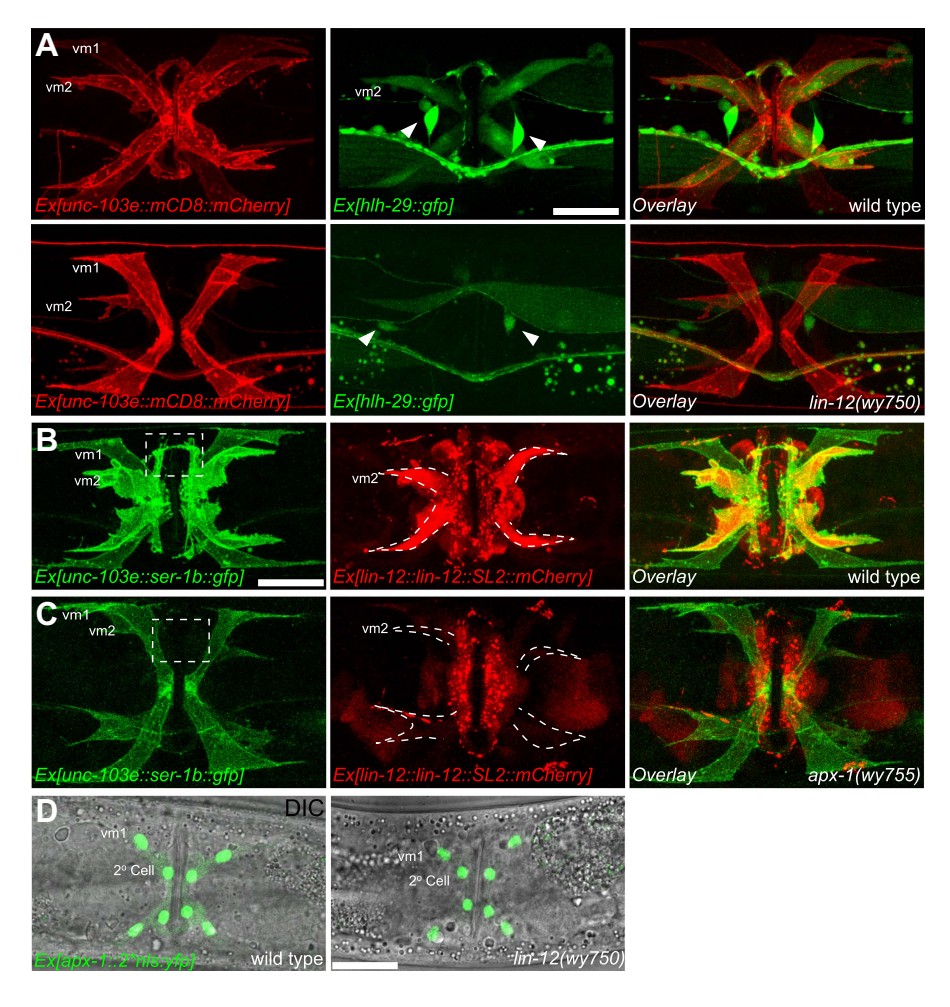

**Figure 5**. vm2 muscle arm development requires APX-1-induced LIN-12/Notch signaling activity. (**A**) *hlh-29*::GFP expression in wild-type (top) and *lin-12(wy750)* (bottom) animals. Left column shows the vulval muscles labeled by mCD8::mCherry. Middle column shows the *hlh-29*::GFP expression patterns. Right column shows the overlaid images. Arrowheads indicate VC4/5 cell bodies. Note that *hlh-29*::GFP is specifically expressed in vm2 but not vm1 in the wild type, and is down-regulated in the *lin-12(wy750)* mutant. Scale bar is 20 μm. (**B**) and (**C**) *lin-12::SL2::mCherry* expression in wild-type (**B**) and *apx-1(wy755)* (**C**) animals. Left column shows the vulval muscles labeled by SER-1::GFP. Middle column shows the *lin-12* expression patterns. Right column shows the overlaid images. Boxed areas indicate synaptic regions. Dashed lines delineate the morphology of vm2. Note that *lin-12* transcription is specifically down-regulated in vm2 cells in *apx-1(wy755)* mutant animals. Scale bar is 20 μm. (**D**) *apx-1* expression in wild-type (left) and *lin-12(wy750)* (right) animals. The four cells in the center of the image are 2° vulval epithelial cells. The four cells at the periphery are vm1 cells. Note that the *apx-1* expression pattern does not change in the *lin-12(wy750)* mutant. Scale bar is 20 μm.

The following figure supplements are available for figure 5:

**Figure supplement 1**. vm2 muscle arm development requires high LIN-12/Notch signaling activity.

postsynaptic muscle arms. The specific activation of the LIN-12/Notch pathway in vm2 but not in vm1 thus appears to dictate the choice of the synaptic target.

We next investigated the mechanism by which *apx-1* activated LIN-12/Notch signaling in vm2. A reciprocal inhibition mechanism as well as a lateral signaling inducing the differentiation of a Notch cell and a Delta cell, mediates the AC/VU decision and VPC specification, respectively (*Greenwald et al., 1983*; *Sternberg, 1988*; *Seydoux and Greenwald, 1989*; *Wilkinson et al., 1994*; *Sundaram, 2004*). We wondered whether a similar mechanism is also used in specifying vm2 identity and regulating

the muscle arm development. We first examined the *lin-12* expression pattern in an *apx-1(wy755)* mutant. Compared with wild-type animals, *apx-1(wy755)* mutant animals showed significantly reduced expression of *lin-12* in vm2 cells, whereas the *lin-12* expression in vulval epithelial cells was not affected (*Figure 5B,C*). On the other hand, we did not observe any change of *apx-1* expression in the *lin-12(wy750)* mutant (*Figure 5D*). These results argue that *apx-1* unilaterally promotes *lin-12* expression and activates the LIN-12/Notch pathway in vm2.

The positive feedback mechanism in the Notch cells is mediated by a CSL induced Notch transcription (*Wilkinson et al., 1994*). Indeed, we found three LAG-1 consensus binding sites in the promoter region of *lin-12* (*Christensen et al., 1996*; *Maier and Gessler, 2000*). To understand if this positive feedback loop is important for vm2 muscle arm formation, we searched for mutations in the *lin-12* promoter region. One *lin-12* loss-of-function allele, *e2621*, was previously known to have egg-laying defects without vulval morphogenesis phenotype (*Wu et al., 1998*). We sequenced the *lin-12* promoter in this allele and found a 541-bp deletion in the promoter region leading to the deletion of two of the three LAG-1 binding sites (*Figure 5—figure supplement 1B*, top). Interestingly, the muscle arm development was similarly affected in *lin-12(e2621)* (*Figure 5—figure supplement 1B*, bottom), suggesting that the muscle arm development might require the positive transcriptional feedback of the LIN-12/Notch pathway.

## *Lin-12* regulates *unc-40* and *madd-2* to promote vm2 muscle arm formation

Given the specific activation of the LIN-12/Notch pathway in vm2 and the cell-autonomous requirement of LIN-12 for muscle arm development, we searched for candidate downstream genes in this pathway. UNC-40, which is a homolog of Deleted in Colorectal Carcinoma (DCC) and neogenin in vertebrates serving as a receptor for the UNC-6/Netrin ligand (*Serafini et al., 1994*; *Chan et al., 1996*; *Keino-Masu et al., 1996*), was reported to play a role in muscle arm formation in the body wall muscles (*Dixon and Roy, 2005*; *Alexander et al., 2009*). More recently, MADD-2, a C1-TRIM protein encoding a homolog of human MID1, was shown to directly bind and function together with UNC-40 to regulate body wall muscle arm extension as well as axon branching and patterning (*Alexander et al., 2010*; *Hao et al., 2010*; *Morikawa et al., 2011*). We therefore examined the vm2 muscle arms in these two mutants. Interestingly, both *unc-40(n324)* and *madd-2(ok2226)* null mutants showed specific vulval muscle arm defects with largely normal vulval muscle morphology (*Figure 6A* and *Figure 1—figure supplement 6*). A significant proportion of the mutant animals completely lacked the vm2 muscle arms, while some other animals showed abnormal muscle arms that extended from incorrect positions or flimsy muscle arms that were thinner and dimmer than the wild-type muscle arms (*Figure 6B*). These defects were strongly rescued by expressing *unc-40* or *madd-2* specifically in vulval muscles in respective mutants, suggesting that both *unc-40* and *madd-2* function cell-autonomously in vm2 to promote and guide muscle arm growth (*Figure 6B*). Consistent with this notion, we found that the expression of both *unc-40* and *madd-2* could be robustly detected in the vm2 cells but not in vm1 cells (*Figure 6C,D*). Moreover, UNC-40::GFP is enriched on the muscle arms, further suggesting that it might function locally to promote muscle arm growth (*Figure 6C*). These results indicate that the levels of UNC-40 and MADD-2 critically determine the capacity of vulval muscles to grow muscle arms and be selected as the postsynaptic target by the egg-laying motoneurons.

To investigate whether *unc-40* and *madd-2* functioned downstream of the LIN-12/Notch pathway, we compared the expression pattern of UNC-40 and MADD-2 in wild-type animals and *lin-12(wy750)* mutants. For these analyses, we used two types of expression constructs: translational reporters and transcriptional reporters. The translational reporters contain the promoter, coding and 3′UTR sequences in which the GFP is inserted in frame with the coding sequences. The fluorescence signal from this type of reporter reflects transcriptional activity, translational regulation and protein stability. The transcriptional reporters only contain the promoter regions and report the transcriptional activity. In wild-type animals, both UNC-40 and MADD-2 translational reporters were preferentially expressed in vm2 cells (*Figure 6C,D*). UNC-40::GFP showed membrane-associated pattern and was enriched on vm2 muscle arms. MADD-2 was distributed diffusely in vm2 cells. The expression of both UNC-40 and MADD-2 in vm2 was significantly down-regulated in the *lin-12(wy750)* mutant (*Figure 6C,D*), supporting the idea that *lin-12* was required for the expression of *unc-40* and *madd-2* in vm2.

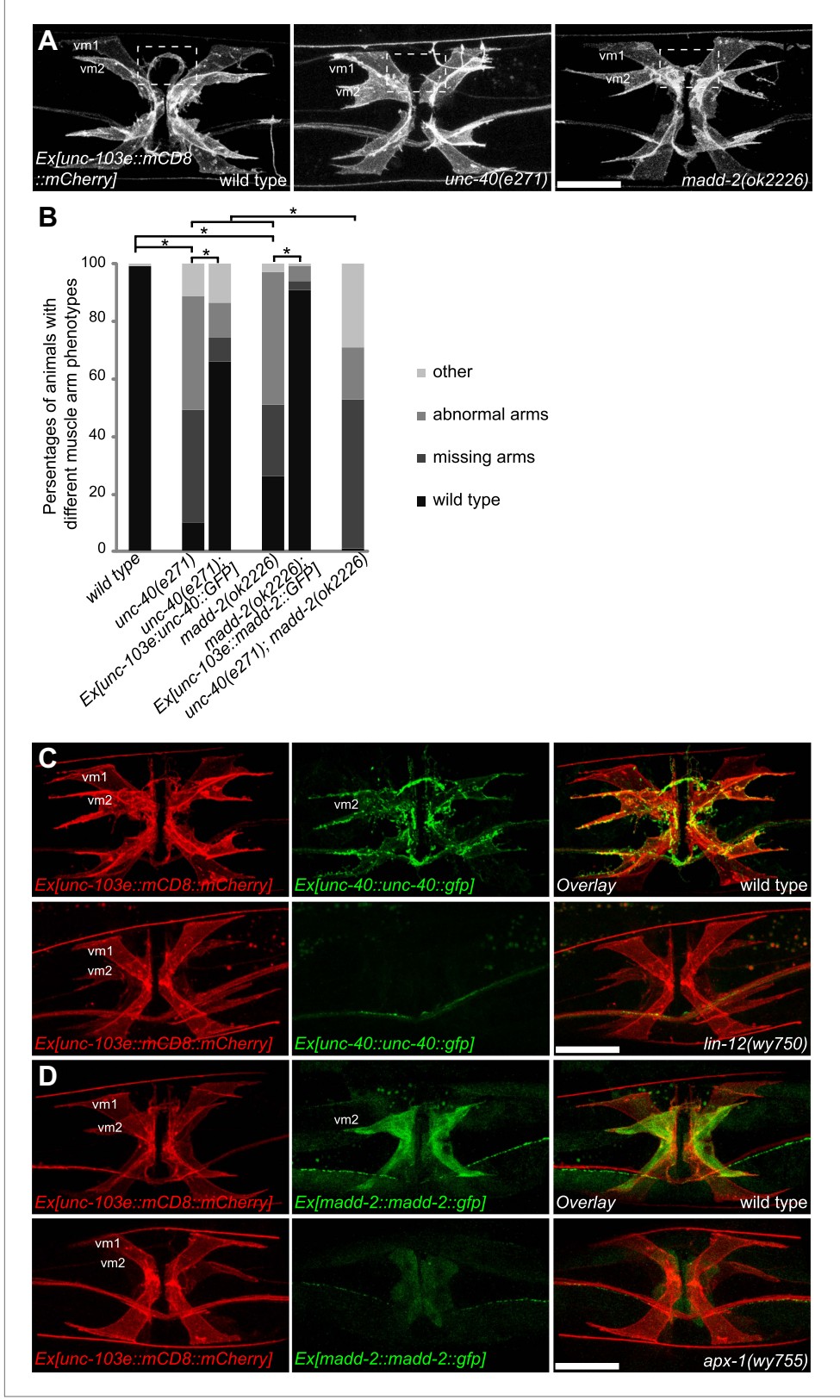

**Figure 6**. LIN-12/Notch signaling instructs vm2 muscle arm by regulating *unc-40/DCC* and *madd-2*.
(**A**) Representative images showing the vulval muscle morphology in wild type (left), *unc-40(e271)* (middle) and *madd-2(ok2226)* (right) animals. Boxed areas indicate synaptic regions. Note that *unc-40(e271)* and *madd-2(ok2226)*
*Figure 6. Continued on next page*

*Figure 6. Continued*

show missing (middle) or abnormal (right) muscle arm phenotypes. Scale bar is 20 μm. (**B**) Quantification of muscle arm phenotypes in animals with different genotypes indicated on the X-axis. 'Wild type' indicates the animals with normal muscle arms. 'Missing arms' indicates the animals with missing muscle arms. 'Abnormal arms' indicates the animals with flimsy muscle arms or abnormal muscle arms extending from incorrect positions. 'Other' indicates animals with severe vulval muscle morphology phenotype that the muscle arms could not be scored. *p<0.0001, chi-squared test, n = 73–194. (**C**) Double-labeling of the expression of UNC-40 and vulval muscle morphology in wild type (top) and *lin-12(wy750)* animals (bottom). Left column shows the vulval muscles labeled by mCD8::mCherry. Middle column shows the UNC-40::GFP expression patterns. Right column shows the overlaid images. Note that UNC-40::GFP is preferentially expressed in vm2 and enriched on the muscle arms in the wild type, and is down-regulated in the *lin-12(wy750)* mutant. Scale bar is 20 μm. (**D**) Double-labeling of the expression of MADD-2 and vulval muscle morphology in wild type (top) and *apx-1(wy755)* animals (bottom). Left column shows the vulval muscles labeled by mCD8::mCherry. Middle column shows the MADD-2::GFP expression patterns. Right column shows the overlaid images. Note that MADD-2::GFP is preferentially expressed in vm2 in the wild type, and is down-regulated in the *apx-1 (wy755)* mutant. Scale bar is 20 μm.

The following figure supplements are available for figure 6:

**Figure supplement 1**. LIN-12/Notch signaling instructs vm2 muscle arm by regulating *unc-40*/DCC and *madd-2*.

To further explore the mechanism by which the expression of *unc-40* and *madd-2* was regulated by *lin-12*, we examined the transcriptional reporters for *unc-40* and *madd-2* to measure the transcriptional activity of these genes. Similar transcription pattern was observed for *unc-40* and *madd-2*. Both *unc-40* and *madd-2* were preferentially transcribed in vm2 cells, and the transcriptional activities of both promoters were dramatically suppressed in the *lin-12(wy750)* mutant (***Figure 6—figure supplement 1A,B***). These data indicate that the canonical LIN-12/Notch signaling pathway acts to promote the transcriptions of both *unc-40* and *madd-2* directly. Consistent with this model, two LAG-1 consensus binding sites were found on the proximal promoter region of the *madd-2* gene. Although no LAG-1 binding site was found in *unc-40* promoter region, two were found in the third intron of *unc-40* that is essential for its expression (***Chan et al., 1996***; ***Christensen et al., 1996***; ***Maier and Gessler, 2000***; Personal communication with Daniel Colon-Ramos, January 2013).

### *unc-40*, potentiated by *madd-2*, determines postsynaptic development

To further examine whether *unc-40* or *madd-2* expression was an output of LIN-12/Notch signaling sufficient for muscle arm formation, we asked whether the overexpression of *unc-40* or *madd-2* with an exogenous promoter in vulval muscles could bypass the requirement of *lin-12*. We found that the expression of *unc-40*, but not *madd-2*, in the *lin-12(wy750)* mutant was sufficient to induced muscle arm-like membrane extension in vm2 cells in ~41% (36 of 88) of the animal (***Figure 7A*** and data not shown), suggestive of a deterministic role of *unc-40* in promoting postsynaptic muscle arm development.

vm1 and vm2 share similar lineage, physical location and morphology. However, only vm2 receives synaptic connections from the egg-laying neurons HSN and VC4/5. Based on the data presented above, we speculated that the LIN-12/Notch signaling pathway elevates the level of UNC-40 and MADD-2 in vm2, which in turn are directly involved in generating muscle arms. To test the sufficiency of UNC-40 and MADD-2 in conferring postsynaptic specificity, we created transgenic animals that expressed UNC-40 ectopically in the vm1 cells. We found that misexpression of UNC-40 was sufficient to induce the dendritic spine-like membrane extensions from vm1 cells (***Figure 7B***). To circumvent the occasional silencing of the UNC-40 transgene, we achieved UNC-40 expression in vm1 by single-copy insertion and observed the similar ectopic muscle arm-like structure (***Figure 7—figure supplement 1***). Although the ectopic membrane extension from vm1 was usually thinner than the wild-type vm2 muscle arm, the vm1 muscle arms always formed from the medial surface of vm1, the identical location as for the normal vm2 muscle arms, suggesting that UNC-40 responds to specific environmental cues to guide the formation of muscle arms.

Similar ectopic expression of *madd-2* in vm1 was not able to induce muscle arm formation (data not shown), hinting that the activity of *unc-40* in vm1 might be required. Since both *unc-40* and *madd-2* mutants showed muscle arm phenotypes, we investigated whether they functioned in the same or parallel pathways. We generated an *unc-40(n324); madd-2(ok2226)* double mutant and found

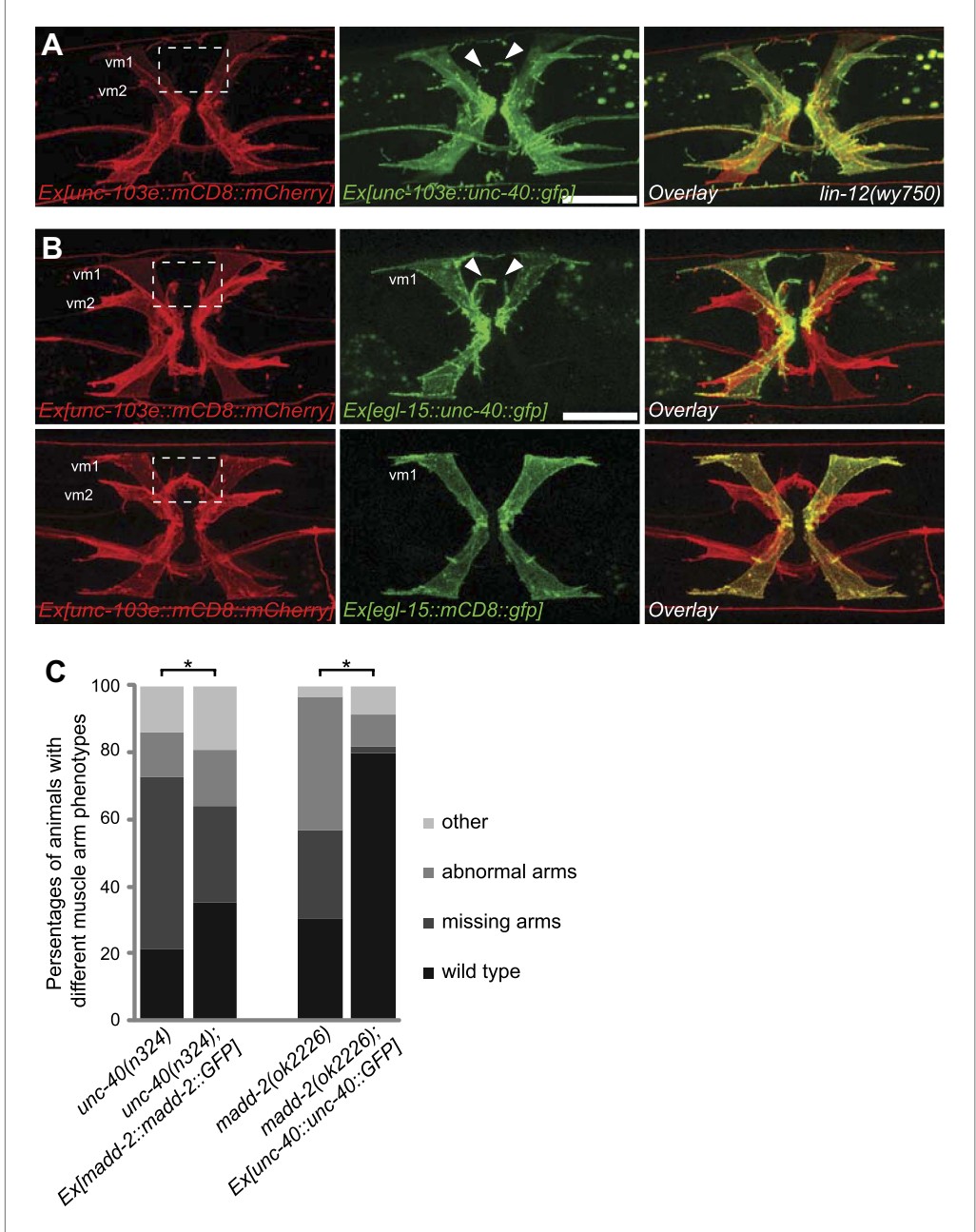

**Figure 7**. *unc-40* and *madd-2* function partly in parallel in regulating vm2 muscle arm development. (**A**) Expression of UNC-40::GFP (middle) in vulval muscles in the *lin-12(wy750)* animals. Vulval muscles are labeled by mCD8::mCherry (left). Boxed area indicates synaptic region. Arrowheads indicate rescued muscle arms. Note that the forced expression of UNC-40 in vm2 from the *unc-103e* promoter induces the formation of muscle arms in the *lin-12(wy750)* mutant. Scale bar is 20 μm. (**B**) Ectopic expression of *unc-40::gfp* (top) or *mCD8::gfp* (bottom) transgene in vm1. Left column shows the vulval muscles labeled by mCD8::mCherry. Middle column shows forced transgene expression patterns. Right column shows the overlaid images. Boxed areas indicate synaptic regions. Arrowheads indicate ectopic muscle arm-like structures. Note that the ectopic expression of UNC-40 in vm1 induces the muscle arm-like membrane extensions. Scale bar is 20 μm. (**C**) Quantification of muscle arm phenotype in animals with different genotypes indicated on the X-axis. 'Wild type' indicates the animals with normal muscle arms. 'Missing arms' indicates the animals with missing muscle arms. 'Abnormal arms' indicates the animals with flimsy muscle arms or abnormal muscle arms extending from incorrect positions. 'Other' indicates animals with severe vulval muscle morphology phenotype that the muscle arms could not be scored. *p<0.0001, chi-squared test, n = 105–311.

*Figure 7. Continued on next page*

*Figure 7. Continued*

The following figure supplements are available for figure 7:

**Figure supplement 1**. Ectopic expression of UNC-40 in vm1 induces muscle arm-like structures.

**Figure supplement 2**. *unc-40* and *madd-2* are not mutually regulated.

**Figure supplement 3**. Schematic model of muscle arm development and vm2 postsynaptic specification.

slightly enhanced muscle arm defects in the double mutants (*Figure 6B*). We also found that the overexpression of *madd-2* partially rescued the muscle arm defects of *unc-40(n324)* null mutant (*Figure 7C*). Conversely, overexpression of *unc-40* largely rescued the muscle arm defects of *madd-2(ok2226)* mutant, suggesting that *madd-2* likely functioned synergistically with *unc-40* to promote muscle arm formation (*Figure 7C*). Expression of *madd-2* and *unc-40* is not mutually regulated (*Figure 7—figure supplement 2A,B*), consistent to the hypothesis that *unc-40* and *madd-2* at least partly function in parallel. We conclude that the postsynaptic development and synaptic target selection in the egg-laying circuit are mediated by the LIN-12/Notch signaling pathway. The secondary vulval epithelial cells and non-target vm1 cells express APX-1/Dsl, which activates its receptor LIN-12/Notch in the vm2 synaptic target cells. The LIN-12/Notch pathway promotes the expression of UNC-40/DCC and MADD-2, which cooperates with each other to generate postsynaptic muscle arms.

## Discussion

Cell type specification likely plays critical roles in synaptic target selection. Our results provide evidence that the canonical LIN-12/Notch pathway, a classic cell fate signaling pathway, determines synaptic specificity by directly controlling the expression of genes that are involved in muscle arm development.

### The vm2 muscle arm development confers the synaptic target selection

The EM reconstruction work proposed that the vm2 muscle arms serve as the postsynaptic specializations, which are similar to the dendritic spines in the vertebrate neurons (*White et al., 1986*). We found that both the postsynaptic potassium channel UNC-103E and a neurotransmitter receptor SER-1B were enriched on vm2 muscle arms, further supporting that the muscle arm structure is functionally relevant postsynaptic specializations (*Figure 1B,C* and *Figure 1—figure supplement 1*). In addition, the mutants lacking vm2 muscle arms show significant defects in the synchronized activation of vm2 and egg-laying behavior (*Figure 2*), indicating the necessity of the vm2 muscle arm for the integrated functional synapse. Therefore, the ability to form muscle arms during development dictates the choice of vm2, but not its sister cell vm1, to become the postsynaptic partner of the egg-laying motor neurons.

### The canonical LIN-12/Notch signaling is required for postsynaptic muscle arm formation at late developmental stage

It has been generally accepted that LIN-12/Notch signaling is involved in multiple cell specification processes. Particularly, LIN-12/Notch signaling is crucial for vulval morphogenesis in *C. elegans* (*Greenwald, 1998*, *2005*). Canonical *lin-12* loss-of-function alleles show defective vulval morphogenesis. However, additional roles of LIN-12/Notch signal have also been suggested because the downregulation of *lin-12* activity at late developmental stage causes defects in the egg-laying behavior without any vulval morphology phenotype (*Sundaram and Greenwald, 1993*). However, little was known about the underlying mechanism. From our study, we definitively showed that the egg-laying defects in *lin-12*, *apx-1*, *sel-12* weak loss-of-function alleles were due to the absence of vm2 muscle arm, which caused asynchronous activation of the anterior and posterior vulval muscles and inefficient egg-laying.

The vm2 muscle arm phenotype in the *lin-12(wy750)* weak allele could be caused directly by the reduction of LIN-12/Notch activity, or indirectly by the disruption of other signaling(s) due to slight alternations of LIN-12/Notch-dependent vulval morphogenesis or other LIN-12/Notch-dependent events during early development. Using a temperature sensitive allele, we showed that disruption of

*lin-12* activity at a very late developmental stage caused vm2 muscle arm defects along with egg-laying defects. We never observed the loss or alternation of vulval muscles in these new weak alleles by examining several cell-type-specific markers (*Figure 1—figure supplement 4C,E,F*), indicating that the function of LIN-12/Notch signaling in specifying the sex myoblasts, the sex muscle precursor cells, is intact (*Greenwald et al., 1983*; *Foehr and Liu, 2008*). This developmental timing matches the relatively late window of vulval muscle specification and muscle arm formation. Furthermore, activation of LIN-12/Notch signaling in vm2 is necessary and sufficient to rescue the vm2 muscle arm development, indicating that LIN-12/Notch signaling is directly required in vm2. These results suggest a novel role of LIN-12/Notch signaling in regulating vm2 postsynaptic muscle arm formation and target selection. Therefore, we conclude that LIN-12/Notch signal instructs both vulval organogenesis and synaptic development of the egg-laying circuit by regulating at least three sequential but independent developmental processes: the AC/VU decision, in which LIN-12/Notch signaling promotes the differentiation of AC and VU cells; VPC specification, in which it determines the vulval precursor cell fates; and vm2 postsynaptic specification, in which it specifies the postsynaptic target cell and further regulates the muscle arm development.

## High activity of LIN-12/Notch signaling may be required to regulate postsynaptic development

While *lin-12*, *apx-1* and *sel-12* null alleles show severe vulval morphogenesis and general vulval muscle morphology phenotypes, all the partial loss-of-function alleles isolated from our screen have specific muscle arm phenotype without other defects. Hence, the muscle arm phenotype shown in *lin-12(wy750)* mutant is not likely the consequence of a complete cell-fate conversion. Even in the muscle lineage, several cell-type specific reporters displayed wild type patterns in *lin-12(wy750)* mutant (*Figure 1—figure supplement 4*), supporting the notion that many aspects of the vm2 differentiation remain intact while only a subset of genes that determines specific postsynaptic development are affected. We can envision two hypotheses to explain the specificity of these alleles. First, vm2 muscle arm development requires a high level of LIN-12/Notch signaling activity, while vulval morphogenesis and overall muscle development require low activity of LIN-12. The new alleles represent partial loss-of-function mutants that specifically disrupt muscle arm development. Alternatively, LIN-12 could execute two different downstream pathways, one required for vulval morphogenesis and the other required for muscle arm development. Since the lesions in almost all the new alleles are point mutations, they might specifically disable the muscle arm development pathway.

We favor the first dosage hypothesis for several reasons. First, the new mutations in *lin-12* and *apx-1* were found in the extracellular domains, but not in the signal transducing cytosolic domain of LIN-12. In addition, a nonsense mutation in *sel-12(wy756, W184stop)* caused strong vulval phenotypes, while a missense mutation in *sel-12(wy760, G373D)* showed specific vm2 muscle arm phenotype, further arguing for the difference in the dosage requirement for these developmental events. Along the same line, a *lag-1* dominant negative construct expressed in the vulval muscles led to muscle arm defects, suggesting that the same canonical LIN-12/Notch signaling pathways is required for the muscle arm development. Second, the *lin-12(e2621)* allele, showed muscle arm defects (*Figure 5—figure supplement 1B*) with intact vulval morphogenesis (*Wu et al., 1998*). Sequencing of the *lin-12* locus of this mutant revealed a 541-bp deletion in the promoter region with no mutation in the coding sequence. The deleted sequence in the promoter region contains two LAG-1 consensus binding sites and therefore is likely to disrupt the positive feedback loop, which could in turn reduce the level of LIN-12/Notch signaling. In fact, the expression of the *hlh-29* transcription reporter was down-regulated in *lin-12(e2621)* (data not shown). Together, these data strongly argue that vulval morphogenesis requires a low level of LIN-12/Notch signaling activation, while vm2 postsynaptic specification requires a relatively high level of LIN-12/Notch activity. Consistent with this model, *mir-61::gfp*, another transcription reporter expressed in VPCs during its specification process, does not change in the *lin-12(wy750)* mutant (data not shown), suggesting that the alternation of LIN-12/Notch signaling activity in *lin-12(wy750)* has little or no effect in VPC specification.

The function of LIN-12 in the AC/VU decision is a case of classic lateral inhibition. A small initial difference in LIN-12/Notch activity between two adjacent cells is amplified by reciprocal feedback mechanisms. The enhanced difference of LIN-12/Notch activity eventually leads to alternative cell fates (*Greenwald et al., 1983*; *Seydoux and Greenwald, 1989*; *Wilkinson et al., 1994*). However, in vm2 muscle arm development, the *lin-12* expression is down-regulated in the *apx-1(wy755)* mutant,

while the *apx-1* expression does not change in the *lin-12(wy750)* mutant. These results indicate a different unilateral induction of LIN-12/Notch signaling activated by APX-1.

## LIN-12/Notch signaling instructs the postsynaptic muscle arm development by up-regulating UNC-40 and MADD-2

The stereotyped vm2 muscle arm structure indicates a specific guidance mechanism regulating the muscle arm extension. Given the fact that *lin-12* is expressed and cell-autonomously required in the postsynaptic target cell vm2, and that the activity of canonical LIN-12/Notch signaling pathway is involved, we speculate that the putative guidance receptors could be the downstream target genes of the canonical LIN-12/Notch pathway. While the canonical LIN-12/Notch signaling has been demonstrated to function in axon guidance and dendrite morphogenesis in fly and mammals (*Berezovska et al., 1999*; *Endo et al., 2007*), its transcriptional targets remain elusive (*Giniger, 2012*).

UNC-40/DCC is a good candidate of the putative guidance receptor. *unc-40/DCC* encodes a *C. elegans* homolog of UNC-6/Netrin receptor (*Serafini et al., 1994*; *Chan et al., 1996*; *Keino-Masu et al., 1996*). It has been suggested to serve as a receptor to guide the growth of axon as well as the extension of dendrite towards the ligand UNC-6/Netrin (*Hedgecock et al., 1990*; *Teichmann and Shen, 2011*). More recently, UNC-40, together with its cytoplasmic activator MADD-2, was shown to regulate body wall muscle arm extension and axon guidance (*Alexander et al., 2010*; *Hao et al., 2010*; *Morikawa et al., 2011*). Indeed, we found that *unc-40* and *madd-2* loss-of-function mutants showed vm2 muscle arm defects, similar to the phenotype found in the *lin-12(wy750)* mutants. Both UNC-40 and MADD-2 are expressed in the postsynaptic vm2 cells but show little or no expression in the non-target vm1 cells. Cell-autonomous rescue of UNC-40 and MADD-2 in the vm2 cells further supports the notion that these two proteins function in vm2 to direct muscle arm development. Interestingly, the expression of the translational reporters for both genes in vm2 is dramatically down-regulated in *lin-12(wy750)* mutant, suggesting that both genes function downstream of LIN-12/Notch signaling pathway.

Consistent to the previous studies in the *C. elegans* body wall muscle arms (*Dixon and Roy, 2005*; *Alexander et al., 2009*), we did not observe muscle arm defects in *unc-6* mutants (data not shown), suggesting the existence of a novel ligand of UNC-40 that functions in guiding the muscle arm extension. The putative ligand provided by vulval epithelial cells could be either a short-range cue that is associated on membrane or locally secreted, or a globally diffusive cue. We favor the former prediction based on the observation that the relative distance between the vulval muscles and the vulval opening is crucial. SM migration is affected in *egl-15(n484)* mutant, which causes posteriorly displaced vulval muscles descended from an SM that stops prematurely during the migration (*Goodman et al., 2003*). Different phenotypes were observed, which appeared to correlate with the location of the vulval muscles. The muscle arms that were located far away from the vulva displayed severe morphogenesis defects, whereas when the vulval muscles did not reach but were still closed to the correct places, we observed muscle arms growing with longer shafts to reach the correct locations defined by HSN presynaptic regions (data not shown). These observations suggest that vulval epithelial cells might provide a membrane-associated or locally secreted cue to guide the vm2 muscle arm extension.

Our additional experiments showed that both *unc-40* and *madd-2* are regulated by LIN-12/Notch signaling at the level of gene transcription. The transcription of *unc-40* and *madd-2* are both specifically activated in vm2 and dependent on *lin-12*, indicating that they could be the direct transcriptional targets of LIN-12/Notch signaling. Consistent with this hypothesis, sequence analysis of *madd-2* promoter and *unc-40* regulatory elements uncovered several LAG-1 consensus binding sites.

Although multiple lines of evidence indicated that UNC-40 and MADD-2 physically interact with each other and function in the same pathway (*Alexander et al., 2010*), the mechanism by which *unc-40* and *madd-2* function is still elusive. In both the Netrin-mediated axon attracting process and Netrin-independent body wall muscle arm extension process, *madd-2* potentiates the activity of *unc-40* (*Hedgecock et al., 1990*; *Dixon and Roy, 2005*; *Alexander et al., 2009*). Consistent to the previous observations, a *madd-2* null allele showed quantitatively less dramatic defects in vm2 muscle arm extension than did *unc-40* or *lin-12* mutants. In addition, we observed only marginally significant enhancement of these muscle arm defects in an *unc-40; madd-2* double mutant. Furthermore, the expression of *unc-40* significantly rescued the vm2 extension phenotype in *madd-2* mutant, while the expression of *madd-2* only showed slight rescue of the *unc-40* mutant. Forced expression of *unc-40* in non-target vm1 cells was

sufficient to induce muscle arms, whereas the overexpression of *madd-2* could not achieve or enhance this phenotype. Taken together, these results demonstrate that *unc-40*, potentiated by *madd-2*, plays a deterministic role in guiding the muscle arm extension and the postsynaptic target selection.

## Specification of the egg-laying circuit requires multiple cell–cell interactions

Our studies of the *C. elegans* egg-laying circuit have yielded surprises in the mechanisms of synapse formation in vivo. Although many in vitro studies suggest that direct interactions between pre- and postsynaptic cells trigger the mutual differentiation processes leading to synaptogenesis, our experiments revealed a significantly different picture. First, the formation of pre- and postsynaptic specializations can occur at different times. The HSN presynaptic specializations form at the early L4 stage, significantly before the maturation of the postsynaptic targets. The extension of the postsynaptic muscle arms takes place 5–6 hr later. Second, the development of the pre- and postsynaptic specializations appears to be independent of each other. Eliminating the postsynaptic cells does not affect the timing and location of synaptic vesicle accumulation in HSN neurons (*Shen and Bargmann, 2003*). Similarly, ablation of the presynaptic neurons has no obvious effect on the timing and direction of the muscle arm growth (*Figure 1—figure supplement 2A,B*). Third, pre- and postsynaptic differentiations are guided by two different but related guidepost cells. Through genetic analysis, we previously found that the primary vulval epithelial cells serve as guidepost cells for the HSN presynaptic development (*Shen and Bargmann, 2003*; *Shen et al., 2004*). These primary cells express a transmembrane IgSF protein SYG-2, which clusters and activates a related molecule, SYG-1 on the HSN axon. The intercellular interaction between SYG-1 and SYG-2 defines the location of HSN synapses and initiates the presynaptic formation (*Shen and Bargmann, 2003*; *Shen et al., 2004*). Interestingly, postsynaptic differentiation is likely guided by the secondary vulval epithelial cells (*Figure 7—figure supplement 3*). APX-1/DSL expressed in the secondary vulval epithelial cells and the non-target vm1 cells activates the LIN-12/Notch signaling pathway in the vm2 cells. Subsequently, UNC-40/DCC and MADD-2 are up-regulated leading to the selective muscle arm formation in vm2. The muscle arms appear to grow along the interface between the primary and the secondary vulval epithelial cells through possibly an UNC-40-dependent mechanism, which ensures that the pre- and postsynaptic compartments will contact each other. The epithelial cells not only function as the channel for egg-laying but also play organizational roles for the development of the entire vulval organ, coordinating multiple events including vulval muscle migration, axon guidance of HSN, VC axon branching and synapse formation. Our discoveries reveal that many of the principles of synaptic organization in vivo might involve cells other than the pre- and postsynaptic neurons.

# Materials and methods

## Strains

Worms were maintained at 20°C on OP50 *Escherichia coli*-seeded nematode growth medium (NGM) plates. N2 Bristol strain worms were used as the wild-type reference. The following mutants were used in this study: *unc-104(e1265) II*, *egl-1(n986) V*, *lin-3(e1417) IV*, *lin-39(n709) III*, *egl-15(n484) X*, *unc-40(e271) I*, *unc-40(n324) I*. The mutants strains *unc-32(e189); lin-12(n676n930) III*, *lin-12(n941)/eT1 (III); him-5(e1467)/eT1[him-5(e1467)] (V)*, *apx-1(or3)/uT1[unc-?(n754);let-?] (IV,V)*, *sel-12(ty11) X*, *dsl-1(ok810) IV*, *glp-1(e2141) III* and *madd-2(ok2226) V* were obtained from Caenorhabditis Genetics Center. *lin-12(e2621) III* was kindly provided by Iva Greenwald.

## Transgenic lines

### Vulval muscle labeling lines

*vsIs147 II (Punc-103e::mCD8::mCherry)*, *vsIs149 V (Punc-103e::mCD8::mCherry)*, *vsSi1 II (Punc-103e::unc-103e::gfp single copy)*, *wyIs321 IV (Punc-103e::unc-103::gfp, Punc-4::mCherry::rab-3)*, *wyIs333 I (Punc-103e::ser-1b::gfp, Punc-86::mCherry::rab-3)*.

### Other markers

*ncIs13 II (Pajm-1::ajm-1::gfp)*, *cyIs4 V (Pcat-1::cat-1::gfp)*, *ayIs2 IV (Pegl-15::gfp)*, *kyIs235 V (Punc-86::snb-1::yfp)*, *leEx1444 (Phlh-29::gfp)*, from CGC, *Phlh-29::gfp III*, kindly provided by Casonya Johnson, *arIs107 (Pmir-61::gfp)*, kindly provided by Iva Greenwald, *ayIs4 I (Pegl-17::gfp)*, *syIs60 II (P(F47B8.6)::gfp)*, *ayIs6 X (Phlh-8::gfp)*, *vsIs4 IV (Prgs-2::gfp)*, kindly provided by Jun Liu.

### Fluorescently tagged protein localization markers

wyEx5200 (Plin-12::lin-12::SL2::mCherry), arIs98 (Papx-1::2*nls::yfp), kindly provided by Iva Greenwald, wyEx604 (Punc-40+minigene::unc-40::gfp), wyEx5657 (Punc-40+minigene::gfp), trIs31 (Pmadd-2::madd-2::gfp), kindly provided by Peter Roy, wyEx5586 (Pmadd-2::mCD8::gfp).

### Protein expression lines

wyEx5402 (Punc-103e::unc-40::gfp), wyEx5658 (Pegl-15::unc-40::gfp), wyEx5281 (Psel-12::sel-12), wyEx5293 (Punc-103e::sel-12 #1), wyEx5294 (Punc-103e::sel-12 #2), wyEx5171 (Papx-1::apx-1), wyEx5197 (Pegl-17::apx-1 #1), wyEx5198 (Pegl-17::apx-1 #2), wyEx5451 (Pegl-15::apx-1 #1), wyEx5452 (Pegl-15::apx-1 #2), wyEx5269 (Punc-103e::dnLag-1 #1), wyEx5270 (Punc-103e::dnLag-1 #2), wyEx5506 (Pegl-15::mCD8::gfp), wyEx5697 (Pegl-17::mCD8::mCherry), wyEx5698 (Punc-103e::madd-2::gfp #1), wyEx5700 (Punc-103e::madd-2::gfp #2).

### Ca$^{2+}$ imaging lines

LX1892 lin-12(wy750) III; lite-1(ce314) X, LX1893 apx-1(wy755) V; lite-1(ce314) X, LX1848 vsIs153 [Punc-103e::GCaMP3, Punc-103e::mCherry, lin-15(+)] IV; lite-1(ce314) X, LX1889 lin-12(wy750) II; vsIs153[Punc-103e::GCaMP3, Punc-103e::mCherry, lin-15(+)] IV; lite-1(ce314) X, LX1890 apx-1(wy755) V; vsIs153[Punc-103e::GCaMP3, Punc-103e::mCherry, lin-15(+)] IV; lite-1(ce314) X

### Genetic screen and SNP mapping

Alleles were isolated from an F2 semi-clonal screen of 3500 haploid genomes using the strain wyIs333. Worms were mutagenized with 50 mM EMS. Complementation test, SNIP-SNP mapping, rescue, and sequencing were performed according to standard protocols.

### Cloning and constructs

All plasmids for transgenic worm lines were made using pPD49.26 vector, ΔpSM or pSM vector, a derivative of pPD49.26 (A. Fire, Stanford).

### vsIs147 II and vsIs149 V

Approximately 2.6-kb DNA fragment upstream of the start site of unc-103e gene was amplified from genomic DNA by PCR using the following oligonucleotides: 5′-GGA ACT AGT TCA TGC CTA TTT TAT ATT TAC AAT ATT TTA G-3′, and 5′-TCA GCG CCC GGG ACC ACC ACC ACC ACA ACC-3′. This fragment was ligated into pPD49.26 to generate pKMC189. mCD8 coding sequence (**Lee and Luo, 1999**) fused to mCherry or GFP was inserted to generate pKMC191 or pKMC192, respectively. pKMC191 was injected at 5 ng/μl with lin-15 rescuing plasmid at 50 ng/μl (**Clark et al., 1994**) into the lin-15(n765ts) animals and then integrated into chromosomes using UV/Trimethylpsoralen. Four independent integrants including vsIs147 and vsIs149 were recovered and backcrossed to the wild type.

### vsSi1 II

unc-103 exons 5 to 11 were PCR-amplified from worm genomic DNA and inserted into the unc-103e cDNA at unique BstXI and AccI sites to generate pKMC100. Coding sequence of GFP was inserted near the 3′ end of unc-103e in a region encoding five consecutive glycine residues to make pKMC179. The unc-103e::gfp DNA fragment was then amplified from pKMC179 by PCR and inserted into the MoSCI vector that derived from pCFJ151 (**Frokjaer-Jensen et al., 2008**) to generate pKMC180. pKMC180 was injected at 50 ng/μl with pJL43.1 encoding the Mos1 transposase at 50 ng/μl and three different counter-selection reporters into EG4322 animals bearing the ttTi5606 Mos1 transposon. vsSi1 was isolated and confirmed by DNA sequencing and expression of GFP.

### wyIs321 IV

pKMC180 at 5 ng/μl and pPL38 (Punc-4::mCherry::rab-3) at 1 ng/μl were injected with Podr-1::dsRed at 25 ng/μl into N2 worms and then integrated into chromosomes using UV/Trimethylpsoralen. wyIs321 was recovered and backcrossed to the wild type six times.

### wyIs333 I

ser-1b cDNA was PCR-amplified from worm cDNA library and subcloned into pKMC192 to replace the mCD8 sequence using the NheI and BamHI sites to generate pPL57. pPL57 at 5 ng/μl and pPL39 (Punc-86::mCherry::rab-3) at 1 ng/μl were injected with Podr-1::dsRed at 25 ng/μl into N2 worms and then integrated into chromosomes using UV/Trimethylpsoralen. wyIs333 was recovered and backcrossed to the wild type six times.

### wyEx5200

SL2::mCherry(kanR+) DNA fragment from pOL007 (*Liu and Shen, 2012*) was amplified and inserted at the 3′ end of *lin-12* genomic sequence using the fosmid recombineering approach (*Tursun et al., 2009*; *Liu and Shen, 2012*) to generate pPL102 (*Plin-12::lin-12::SL2::mCherry*). pPL102 was injected at 5 ng/µl with *Podr-1::dsRed* at 25 ng/µl into *lin-12(wy750); wyIs333* worms.

### wyEx5281

*sel-12* cDNA was PCR-amplified from worm cDNA library and ligated into ΔpSM using AscI and SalI sites to generate pPL154. Approximately 2.5-kb DNA fragment upstream of the start site of *sel-12* gene was amplified from worm genomic DNA by PCR using the following oligonucleotides: 5′-ACG CGT CGA CGC ATA TAA TAA ACA CTT TTG AGA GAC TTG TG-3′, and 5′-AAG GAA AAA AGC GGC CGC GTA TTC ACA CGA GTA CAG TCT GA-3′. This DNA fragment was then inserted into pPL154 using NotI and AscI sites to generate pPL155 (*Psel-12::sel-12*). pPL155 at 15 ng/µl was injected with *Podr-1::dsRed* at 25 ng/µl into *sel-12(wy760); wyIs333* worms.

### wyEx5293 and wyEx5294

*unc-103e* promoter from pKMC192 was subcloned into pPL154 using NotI and AscI sites to generate pPL157 (*Punc-103e::sel-12*). pPL157 at 5 ng/µl was injected with *Podr-1::gfp* at 25 ng/µl into *sel-12(wy760); wyIs333* worms. Two independent extrachromosomal arrays were recovered.

### wyEx5171

*apx-1* cDNA was PCR-amplified from worm cDNA library and ligated into ΔpSM using AscI and XmaI sites to generate pPL118. Approximately 4-kb DNA fragment upstream of the start site of *apx-1* gene was amplified from worm genomic DNA by PCR using the following oligonucleotides: 5′- AAG GAA AAA AGC GGC CGC GAT TTG AAT ATT TAT TAG TTT GCC AGG T-3′, and 5′- TTG GCG CGC CAG TAC AGG ATC GTG TGC TAG A-3′. This DNA fragment was then inserted into pPL118 using NotI and AscI sites to generate pPL120 (*Papx-1::apx-1*). pPL120 at 10 ng/µl was injected with *Podr-1::gfp* at 25 ng/µl into *apx-1(wy755); wyIs333* worms.

### wyEx5197 and wyEx5198

Approximately 4-kb DNA fragment upstream of the start site of *egl-17* gene was amplified from worm genomic DNA by PCR using the following oligonucleotides: 5′-ACA TGC ATG CCT ATG CAG CAT TGG AGG ATG ATG-3′, and 5′-TTG GCG CGC CAG CTC ACA TTT CGG GCA CCT GAA-3′. This DNA fragment was then inserted into pPL118 using NotI and AscI sites to generate pPL135 (*Pegl-17::apx-1*). pPL135 at 15 ng/µl was injected with *Podr-1::gfp* at 25 ng/µl into *apx-1(wy755); wyIs333* worms. Two independent extrachromosomal arrays were recovered.

### wyEx5451 and wyEx5452

Approximately 1-kb DNA fragment upstream of the start site of *egl-15* gene was amplified from worm genomic DNA by PCR using the following oligonucleotides: 5′-ACA TGC ATG CAA TCG CAC AGA TTA AAT TAC ACT TC-3′, and 5′-TTG GCG CGC CGA TGA TAT GTA TCA GGC AAA CAT TC-3′. This DNA fragment was then inserted into pPL118 using NotI and AscI sites to generate pPL194 (*Pegl-15::apx-1*). pPL194 at 15 ng/µl was injected with *Podr-1::gfp* at 25 ng/µl into *apx-1(wy755); wyIs333* worms. Two independent extrachromosomal arrays were recovered.

### wyEx5269 and wyEx5270

*lag-1* cDNA was PCR-amplified from worm cDNA library. A g to a single nucleotide mutation was made by overlapping PCR to generate R397H alternation. The product was confirmed by sequencing and then inserted into ΔpSM using AscI and SalI sites to generate pPL141. *unc-103e* promoter from pKMC192 was subcloned into pPL141 using SphI and AscI sites to generate pPL145 (*Punc-103e::dnLag-1*). pPL145 at 30 ng/µl was injected with *Podr-1::gfp* at 25 ng/µl into *wyIs333* worms. Two independent extrachromosomal arrays were recovered.

### wyEx5506

*egl-15* promoter from pPL194 was subcloned into pKMC192 to replace the *unc-103e* promoter and generate pPL213 (*Pegl-15::mCD8::gfp*). pPL213 at 5 ng/µl was injected with *Podr-1::DsRed* at 25 ng/µl into *vsIs147* worms.

### wyEx5697

*egl-17* promoter from pPL135 was subcloned into pKMC191 to replace the *unc-103e* promoter and generate pPL247 (*Pegl-17::mCD8::mCherry*). pPL247 at 10 ng/µl was injected with *Podr-1::DsRed* at 25 ng/µl into *arIs98* worms.

### wyEx5402

*unc-103e* promoter from pKMC192 was subcloned into pHMT15 (*Teichmann and Shen, 2011*) using SphI and AscI sites to replace the *mig-13* promoter and generate pPL180 (*Punc-103e::unc-40::gfp*). pPL180 at 15 ng/µl was injected with *Podr-1::DsRed* at 25 ng/µl into *vsIs147* worms.

### wyEx5658

*egl-15* promoter from pPL194 was subcloned into pHMT15 (*Teichmann and Shen, 2011*) using SphI and AscI sites to replace the *mig-13* promoter and generate pPL185 (*Pegl-15::unc-40::gfp*). pPL185 at 10 ng/µl was injected with *Podr-1::DsRed* at 25 ng/µl into *vsIs147* worms.

### wyEx5657

The regulatory region containing ~0.7 kb of sequence 5′ upstream to the initiator methionine in exon 1, with exons 1–5, introns 1–5 (total ~6.3 kb) of *unc-40* gene was amplified from worm genomic DNA by PCR using the following oligonucleotides: 5-AAG GAA AAA AGC GGC CGC GGG TTC TGA AAT AGA AAA AAG TTG TTT CG-3′, and 5′-TTG GCG CGC CTG CCT CTG GTC TAG CAG CCG CC-3′. This DNA fragment was then inserted into pSM using NotI and AscI sites to generate pPL309 (*Punc-40+minigene::gfp*). pPL309 at 5 ng/µl was injected with *Podr-1::DsRed* at 25 ng/µl into *vsIs147* worms.

### wyEx5698 and wyEx5700

*madd-2* cDNA was PCR-amplified from worm cDNA library and inserted into pSM using AscI and XmaI sites to generate pPL197. *unc-103e* promoter from pKMC192 was subcloned into pPL197 using SphI and AscI sites to generate pPL201 (*Punc-103e::madd-2::gfp*). pPL201 at 50 ng/µl was injected with *Podr-1::DsRed* at 25 ng/µl into *vsIs147* worms. Two independent extrachromosomal arrays were recovered.

### wyEx5586

Approximately 2.5-kb DNA fragment upstream of the start site of *madd-2* gene was amplified from worm genomic DNA by PCR using the following oligonucleotides: 5′-ACA TGC ATG CTG CAA GTC TAA TAG GGA GGC-3′, and 5′-TTG GCG CGC CTT TTG AAG GGA CTG AAA TTT GAG T-3′. This DNA fragment was then inserted into pKMC192 using SphI and AscI sites to replace the *unc-103e* promoter and generate pPL219 (*Pmadd-2::mCD8::gfp*). pPL219 at 15 ng/µl was injected with *Podr-1::DsRed* at 25 ng/µl into *vsIs147* worms.

## Fluorescence confocal imaging and quantifications

Fluorescence images were captured in live *C. elegans* using a Plan-Apochromat 63 3 1.4 objective on a Zeiss LSM710 confocal microscope. Worms were immobilized using a mixture of 200 mM 2,3-butanedione monoxime (Sigma-Aldrich, St Louis, MO) and 2.5 mM levamisole (Sigma-Aldrich) in 10 mM Na HEPES.

For quantifying the vulval muscle sizes and the intersection angles between vulval slit and vulval muscles, 12–16 confocal images were used for each genotype. Each vulval muscle was lineated by polygonal lines in ImageJ to obtain the size. The middle points of the connection lines where vm1 was linked to the body wall, and the tips of vm2 cells were used as the landmarks to measure the intersection angles.

## Egg-laying assay

Twenty L4-stage worms of each genotype were isolated and allowed to develop 30 hr at 20°C into gravid adults. The worms were transferred to a fresh plate and allowed to lay eggs for 2 hr at 20°C. The adult worms were removed before the total number of each plate and the stage of each egg were scored. The eggs were classified into one of three developmental stages: 1–8 cell, eight-cell to comma, or post-comma. Three independent assays were performed with at least 50 eggs quantified in total for each genotype. Fisher's exact test was performed to compare statistical significance between two groups.

## Ca²⁺ imaging

To perform Ca²⁺ imaging in the vulval muscles, we used as a control strain LX1890 *vsls153; lite-1(ce314)*, which expresses GCaMP3 and mCherry from the *unc-103e* promoter and is blue light insensitive, as previously described (*Collins and Koelle, 2012*). KG1180 animals carrying the *lite-1(ce314)* mutation (*Edwards et al., 2008*) were crossed to *lin-12(wy750)* or *apx-1(wy755)* animals to generate LX1892 *lin-12(wy750); lite-1(ce314)* and LX1893 *apx-1(wy755); lite-1(ce314)*, respectively. LX1890 *vsls153; lite-1(ce314)* animals were crossed with LX1892 *lin-12(wy750); lite-1(ce314)* and LX1893 *apx-1(wy755); lite-1(ce314)* to generate LX1889 *lin-12(wy750); vsls153; lite-1(ce314)* and LX1890 *vsls153; apx-1(wy755); lite-1(ce314)*.

Ratiometric Ca²⁺ imaging of behaving animals was performed as previously described (*Collins and Koelle, 2012*). Single worms aged 24 hr after the L4 stage were picked with a small amount of OP50 bacteria to an unseeded NGM plate. From this plate, a ~20 × 20-mm chunk was placed worms-side down onto a 24 × 60 mm #1 coverslip and overlaid with a 22 × 22 mm #1 coverslip. The worms were allowed to recover for 1–2 hr in a humidified chamber at 20°C before imaging. Two-channel confocal slices (18-μm-thick, 256 × 256 pixel, 16-bit) were collected for 6 min at 20 Hz through a 20× Plan-Apochromat objective (0.8NA) using the LIVE detector of an inverted Zeiss 710 Duo confocal microscope. The stage and focus were adjusted manually to keep the egg-laying system in view and focused during recording periods. Ratiometric analysis was performed in Volocity (version 5.4; Perkin Elmer). Briefly, separate GCaMP3/mCherry ratio and intensity-modulated ratio channels were calculated. Voxels with mCherry fluorescence intensities 2 standard deviations above background were selected as objects for measurement (typically ~500 voxels per time point). The average ratio of these voxels were smoothened using a 150 ms (three time points) rolling average, and the lowest 10% of the GCaMP3/mCherry ratio values were averaged to establish a ΔR/R (change in GCaMP3/mCherry fluorescence ratio/absolute ratio) baseline. Ca²⁺ transients were identified by visual inspection of intensity-modulated ratio movies and ratio traces, and ΔR/R peaks above 15% were analyzed. Transients were classified as single when the transient was evidently confined to the anterior or posterior vulval muscle and double when both anterior and posterior muscles had obvious transients. Double transients that resolved to a single peak in the ratiometric trace were defined as having no delay (0 s, *Figure 2K*). When two or more transients within a single body bend were separable both spatially and temporally in the intensity-modulated ratio movies and had independent peaks separated by ≥250 ms in the ratiometric trace, the delay between the anterior and posterior peaks was determined.

## Acknowledgements

We thank the international *C. elegans* Gene Knockout Consortium, the National Bioresourse Project (Japan), and the laboratories of P Roy, C Johnson, and K Liu for strains. We also thank C Gao and B Tara for technical assistance. We thank I Greenwald for sharing unpublished results, reagents and thoughtful comments throughout this project.

## Additional information

### Funding

| Funder | Grant reference number | Author |
| --- | --- | --- |
| Howard Hughes Medical Institute | | Kang Shen |
| American Heart Association | POST4990016 | Kevin M Collins |
| National Institute of Neurological Disorders and Stroke | NS036918 | Michael R Koelle |
| Yale Liver Center | DK34989 | Michael R Koelle |

The funders had no role in study design, data collection and interpretation, or the decision to submit the work for publication.

### Author contributions

PL, Isolated and identified the mutants, characterized the mutant phenotype and performed the genetic interaction experiments, wrote the paper; KMC, MRK, Performed the behavioral and physiological experiments; KS, Supervised the project, wrote the paper.

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
