## [Decision Letter]

Thank you for choosing to send your work entitled “LIN-12/Notch signaling instructs postsynaptic target selection by regulating UNC-40/DCC and MADD-2 in *C. elegans*” for consideration at *eLife*. Your article has been favorably evaluated by a Senior editor and 3 reviewers, one of whom is a member of our Board of Reviewing Editors.

The Reviewing editor and the other reviewers discussed their comments before we reached this decision, and the Reviewing editor has assembled the following comments to help you prepare a revision.

1. The major concern of all three reviewers centers on the interpretation of the muscle arm phenotype as a defect in synapse specificity. The data do not address a “synaptic function” of Notch in this system (as, for example, in the studies of Alberi et al. or de Bivort et al.). However, the data do document an interesting muscle differentiation program that precedes postsynaptic differentiation and the participation of the muscle cell in synapse development through muscle arm formation. An appropriate interpretation should focus on the muscle arm phenotype, the developmental program for muscle arm formation controlled by Notch, and the involvement of these processes in synapse development. However, the concept of synapse specificity, highlighted in the title and throughout the text, is an interpretation that is not directly addressed by the data. The text and title should be substantially revised to address this issue.

2. There was discussion among the reviewers regarding whether the absence of Notch signaling in vm2 partially switches fate or has a more selective disruption of the muscle arm growth program. The analysis of more fate markers and a more detailed description of morphological features of the vm2 muscle (e.g., the positioning of its attachment sites) in *lin-12* mutants could clarify this issue. In any case, this needs to be more directly addressed in the Discussion.

---

## [Author Response]

*1. The major concern of all three reviewers centers on the interpretation of the muscle arm phenotype as a defect in synapse specificity. The data do not address a “synaptic function” of Notch in this system (as, for example, in the studies of Alberi et al. or deBivort et al.). However, the data do document an interesting muscle differentiation program that precedes postsynaptic differentiation and the participation of the muscle cell in synapse development through muscle arm formation. An appropriate interpretation should focus on the muscle arm phenotype, the developmental program for muscle arm formation controlled by Notch, and the involvement of these processes in synapse development. However, the concept of synapse specificity, highlighted in the title and throughout the text, is an interpretation that is not directly addressed by the data. The text and title should be substantially revised to address this issue*.

We agree with the reviewers that our data do not address the role of the Notch in synaptic function, similar to the reports by Alberi et al. and de Bivort et al. In those two studies, the authors documented the function of the Notch pathway in synaptic plasticity in the mature nervous system. In our manuscript, we are studying the role of Notch during the development of the neural circuit.

We also agree with the reviewers that we are studying a “muscle differentiation program that precedes postsynaptic differentiation and the participation of the muscle cell in synapse development through muscle arm formation”. We have made changes in the title and throughout the text to more precisely reflect the specific function in muscle arm development.

However, we do feel that our data strongly indicate that the Notch-mediated muscle arm formation is a critical determinant of target specificity in the egg-laying circuit. We started the project with a forward genetic screen on a marker that labeled the postsynaptic specializations on the muscle arms. In the partial loss-of-function mutant alleles of the Notch pathway that we isolated in this synaptic morphology screen, many aspects of the muscle differentiation remain intact, suggesting that high level of the Notch signaling activity is specifically required for the postsynaptic development. In other words, the vm2 cells absolutely require high level of Notch signaling to become the postsynaptic target of the presynaptic HSN neurons.

The next question is how LIN-12 affects muscle arm target selection. We found that LIN-12 activity is only found in vm2 but not vm1. Furthermore, LIN-12 acts cell-autonomously to activate the expression of UNC-40/DCC and MADD-2 in vm2. These two molecules in turn are required for vm2 muscle arm formation and targeting. Most importantly, to demonstrate that the LIN-12/Notch-controlled molecular program is directly involved in target selection during neural circuit assembly, we showed that UNC-40 is sufficient to induce muscle arm-like membrane protrusions when misexpressed in vm1. Taken together, these data suggest that the high-level activity of LIN-12 distinguishes vm2 from vm1 in the process of forming muscle arms and becoming the postsynaptic targets in the egg-laying circuit.

*2. There was discussion among the reviewers regarding whether the absence of Notch signaling in vm2 partially switches fate or has a more selective disruption of the muscle arm growth program. The analysis of more fate markers and a more detailed description of morphological features of the vm2 muscle (e.g., the positioning of its attachment sites) in lin-12 mutants could clarify this issue. In any case, this needs to be more directly addressed in the Discussion*.

We thank the reviewers for the suggestions. We have performed several additional experiments. In the revised manuscript, besides the vm1 specific marker *egl-15::gfp* (Figure 1–figure supplement 4C), we have included analyses of two additional cell-type specific markers (*hlh-8::gfp* and *rgs-2::gfp)* (Figure 1– figure supplement 4E and 4F). *hlh-8::gfp* labels the M lineage, which include vm1, vm2, and the uterine muscles. This marker is only present in differentiating muscles and is absent in mature muscles. *rgs-2::gfp* is a specific marker for the uterine muscle. In the *lin-12(wy750)* mutant animals, the expression of *hlh-8::gfp* is found in all three classes of muscles only during L3-L4, indistinguishable from the wild type controls. Similarly, in the *lin-12(wy750)* mutant animals, the expression of *rgs-2::gfp* is found only in the uterine muscles, indistinguishable from the wild type controls. Together, we have examined three differentiation markers for the vm2 and related muscle lineage. We found no obvious gross cell fate changes in the *lin-12(wy750)* mutant. We want to remind the reviewers that this *lin-12* allele is a partial loss-of-function allele, which showed no vulval morphogenesis phenotypes (another cell fate phenotype). We think the lack of cell fate phenotype is likely due to the remaining LIN-12/Notch activity in this strain.

We have also followed the reviewers' suggestion and characterized the morphology of the vulval muscles quantitatively. The results are now presented in Figure 1–figure supplements 5 and 6. We show that the general vulval muscle morphology is not affected in *lin-12(wy750)* and *unc-40(n324)* mutants, further supporting the notion that the phenotype of *lin-12(wy750)* is restricted on the muscle arm structure.